# Demystifying Latent Forgetting in Federated Learning

## Abstract

Federated Learning (FL) enables collaborative model training across decentralized, isolated clients in a privacy-preserving manner, but at the cost of limited control over the data and the training procedure. One of the key challenges in FL is the *spatial* data heterogeneity, which is due to the stratified nature of the underlying data distributions between clients. In addition, FL systems also undergo periods of time in which certain features disappear from the training data pool, resulting in the less studied but critical problem of *temporal-spatial* data heterogeneity. Such non-uniformity in training data across time introduces a new feature-level latent forgetting that is fundamentally different from the well-studied task-level catastrophic forgetting in continual learning. This latent forgetting, if not detected and mitigated timely, can result in poor model performance, especially for certain learning features. The privacy requirements and temporal-spatial data heterogeneity of FL make the detection and mitigation of latent forgetting challenging. In this paper, we analyze latent forgetting and propose FedMemo, a privacy-preserving FL framework to control its impact. FedMemo employs an automated detection mechanism to detect latent forgetting in real time with preserved privacy. FedMemo further introduces a proxy-based 2-step aggregation approach to mitigate the impact of latent forgetting. We evaluate FedMemo in a diverse set of vision and language classification tasks in various FL settings, and show that it outperforms state-of-the-art methods by up to $20.06\%$ in CIFAR-10 and SVHN, and up to $25.69\%$ in GLUE, effectively mitigating the challenges posed by temporal-spatial data heterogeneity.

## 1 Introduction

Federated learning (FL) McMahan et al. (2017) enables collaborative model training across distributed devices while preserving the privacy of local data, which makes it attractive in edge computing Voigt & Von dem Bussche (2017). In a classic learning process of FL, clients perform local training and send model updates to a central server. The server aggregates the updates and broadcasts the updated global model to clients. This process maintains data privacy as the server never accesses clients' raw data, labels, or distribution details. There is no direct information exchange among clients either. Although FL addresses the privacy issue, it introduces several practical training challenges. One well-known major problem is *spatial data heterogeneity* Kairouz et al. (2021) Li et al. (2020), where client data is not independent and identically distributed (non-IID) due to variations in feature distributions, label proportions, or local environments resulting in divergent local updates that impairs global model convergence McMahan et al. (2017); Shoham et al. (2019); Xu et al. (2022); Kairouz et al. (2021); Li et al. (2020).

In the real-world FL systems, the training process may encounter dynamic and unpredictable environments due to intermittent client connectivity, shifting data availability, and dynamic data content. These changes could result in a shift in the trainable features available at different training periods. For example, weather sensors can go months without observing snowfall, smartphones may run out of power or go out of network. Such non-stationarity feature availability leads to instability in optimization, due to the stochastic gradients computed on different clients become biased estimators of the global gradient, resulting in model drift, slower convergence, and degraded generalization performance Xu et al. (2022); Capanema et al. (2025). Such trainable feature shift essentially introduces another dimension of heterogeneity, which we name *temporal data heterogeneity*. With *temporal data heterogeneity*, the model must adapt to the current available feature, while also maintaining its performance on the temporally or permanently unavailable features to avoid forgetting the already learned features Kirkpatrick et al. (2017); McCloskey & Cohen (1989). Combined with *spatial data heterogeneity*, we call the heterogeneity that exists in both a particular training round and across training rounds *temporal-spatial data heterogeneity*. Such non-uniformity in training data across

clients and training rounds introduces a new feature-level forgetting, which can result in poor model performance if not detected and mitigated timely.

It is important to point out that in traditional centralized learning systems, the challenges in training due to data shift have been extensively studied in the field of *continual learning* (CL) Li & Hoiem (2017); Li et al. (2025) (also referred to as *lifelong learning* or *incremental learning* Aljundi et al. (2017); Castro et al. (2018)). Specifically, such a loss in learned features is called catastrophic forgetting, which occurs when models overwrite previously acquired knowledge after training on new task McCloskey & Cohen (1989); Kirkpatrick et al. (2017). However, in regular FL training, the task is the same, the temporally or permanently unavailable features during training is usually unexpected and uncontrollable. In addition, methods focusing on centralized training such as Aljahdali et al. (2024); Bhope et al. (2025); Li et al. (2025) often assume access to task boundaries, sequential task ordering, task-specific metadata, assumptions that rarely hold in FL environment.

To clearly distinguish from the catastrophic forgetting studied in the traditional continual learning systems, we denote the forgetting in FL system *latent forgetting*, which occurs due to the *temporal-spatial data heterogeneity* we discussed earlier. We further categorize *latent forgetting* into two types: *short-term latent forgetting*, where feature segments are missing temporarily (e.g., due to seasonal trends), and *long-term latent forgetting*, where some features never show up in the rest of training. Unlike the catastrophic forgetting, which is relatively easy to detect as it usually happens during task switching. The *latent forgetting* in FL is hard to detect due to the privacy requirement that prevents measuring the changes in data or feature distribution across training.

In this paper, we introduce a FL framework FedMemo for effectively detecting and mitigating latent forgetting in FL without violating privacy. We first introduce a privacy-preserving metric for detecting latent forgetting in Federated Learning (FL), which only uses global model weight updates without requiring access to private client information. We then leverage synthetic proxy data generated via a server-side GAN, or optionally via a generative model on clients, to preserve previously learned knowledge. We further design a novel two-step aggregation method to effectively incorporate proxy gradients to mitigate latent forgetting. We formulate *temporal-spatial data heterogeneity* and conduct theoretical analysis to show how latent forgetting would result in poor performance and how our 2-step aggregation with synthetic proxy data can mitigate such impact.

Through extensive evaluations, we demonstrate that on CIFAR-10 and SVHN, FedMemo outperforms a strong baseline method that we adapted from state-of-the-art Federated Class Incremental Learning (FCL) by up to 20.06%. On GLUE benchmarks, FedMemo achieves a 25.69% better average accuracy compared to the best existing FL aggregation methods, while still maintaining the same communication overhead and incurring minimal additional computation cost.

## 2 RELATED WORKS

**General methods for catastrophic forgetting -** Many studies have tackled catastrophic forgetting in neural networks, such as EWC Kirkpatrick et al. (2017), which emphasizes important weights for previous tasks but assumes well-defined task-level data distributions. Memory-based methods Aljundi et al. (2017) also show success in preserving knowledge across tasks, but they require storing raw or replay data, raising privacy concerns in FL. In FL, studies like Shoham et al. (2019); Xu et al. (2022) partition clients based on classes but limit participation to a few consecutive rounds, which do not address the dynamic and periodic participation typical of real-world FL.

**Distillation-based methods for mitigating forgetting -** Paper Lee et al. (2022) addresses forgetting by distilling global knowledge from client models regarding "not-true" classes, but introduces a hyperparameter to balance new knowledge acquisition with preserving previous knowledge, without considering the risk of overfitting in high heterogeneity levels. Dong et al. (2023) proposes LGA, which balances local class imbalances using gradient-adaptive compensation and semantic distillation losses, while employing a proxy server to collect perturbed images from clients in FCL. However, synthetic data in such methods poses privacy risks as received from clients. Lin et al. (2020); Sattler et al. (2021); Seo et al. (2022) use unlabeled proxy data to aggregate models from local clients with different architectures, which differs from our approach. Recent paper Aljahdali et al. (2024) addresses forgetting by having each local client distill knowledge from the previous global model, but it only considers spatial data heterogeneity and does not account for data shifts caused by temporal-spatial heterogeneity. The state-of-the-art methods for FCL, Babakniya et al. (2023), proposes MFCL

which uses data-free knowledge distillation Haroush et al. (2020), but it assumes predefined task-level uniform data distributions, and fixed task transitions, which are not realistic for FL settings. It is also based on accuracy decay, which can be influenced by various factors like overfitting or tuning and is only explored on vision tasks. We compare FedMemo against MFCL customized for FL, and demonstrate that our approach consistently outperforms it.

**GAN-based methods for mitigating forgetting -** Methods like FeGAN Guerraoui et al. (2020), FedGAN Rasouli et al. (2020), and MD-GAN Hardy et al. (2019) leverage GANs to address spatial data heterogeneity and reduce well-known catastrophic forgetting in federated learning. however, they push significant computational or communication burdens on clients. CAP Zhang et al. (2023) introduces device-to-edge communication, requiring additional infrastructure and protocol complexity beyond the standard FL setup. Some works Li et al. (2022) rely on clients uploading locally generated synthetic data to form a shared global dataset, but this approach scales poorly and incurs high resource and communication costs. Despite these efforts, existing methods largely ignore latent forgetting in FL caused by temporal-spatial heterogeneity—a critical gap we aim to address.

All the existing works only focus on conventional spatial heterogeneity while overlooking the temporal-spatial heterogeneity impact in practice. We address this issue by introducing a temporal-spatial heterogeneity aware approach.

## 3 FL LATENT FORGETTING DETECTION AND MITIGATION

### 3.1 PROBLEM FORMULATION AND NOTATIONS

We consider a standard FL setup with K clients. Each client $i$ has a private data distribution $P_i^{(t)}(x, y)$ at round $t$, where $x \in \mathcal{X}$ denotes an input from the feature space $\mathcal{X}$ and $y \in \mathcal{Y} = \{1, \dots, C\}$ denotes the corresponding class label. The model is represented by a function $f_w : \mathcal{X} \rightarrow \mathcal{S}$ parameterized by $w$, where $\mathcal{S} = \{(s_1, \dots, s_C) \in \mathbb{R}^C \mid s_j \geq 0, \sum_{j=1}^C s_j = 1\}$ is the probability simplex over the $C$ classes. For an input $x$, the model outputs a vector of class probabilities in $\mathcal{S}$. The loss function of client $i$ Karimireddy et al. (2020); Zhao et al. (2018) is defined as: $\mathcal{L}_i^{(t)}(w) = \mathbb{E}_{(x,y) \sim P_i^{(t)}} \left[ \ell(f_w(x; w), y) \right]$ where $\ell(.,.)$ is the per-sample loss function (e.g. cross entropy loss). The server aggregates local updates from the participating clients $K_t \subseteq \{1, \dots, K\}$ in round $t$ using weighted averaging: $w^{t+1} = \sum_{i \in K_t} w_i^{t+1} \cdot \frac{N_i}{\sum_{j \in K_t} N_j}$ where $N_i$ is the number of samples held by client $i$, and $w_i^{t+1}$ denotes its updated local model parameters McMahan et al. (2017). We next define following heterogeneities.

**Spatial data heterogeneity** arises when different clients have non-identical distributions: $P_i^t(x, y) \neq P_j^t(x, y)$, for some clients $i \neq j$, Note that unlike most federated continual learning studies where spatial heterogeneity is considered only as the conditional feature distributions $P_i^t(x \mid y) \neq P_j^t(x \mid y)$ (such as Huang et al. (2022)), FL also consists of labels skew $P_i(y)^t \neq P_j^t(y)$

**Temporal data heterogeneity** occurs when the distribution of a single client evolves over rounds, causing seasonal shifts, feature disappearance, or gradual drifts, where $P_i^{(t)}(x, y) \neq P_i^{(t-1)}(x, y)$.

**Temporal-spatial heterogeneity** derives from both spatial and temporal heterogeneities where $P_i^{(t)}(x, y) \neq P_j^{(t')}(x, y), (i, t) \neq (j, t')$. The above heterogeneities impact how the global model is trained. Spatial heterogeneity implies that some clients may never observe parts of the data. By temporal heterogeneity, some samples or features may disappear within the same client over rounds. Together, temporal-spatial heterogeneity causes under-representation in the aggregated updates, leading to the global model's gradual loss of sensitivity to infrequent features, even without abrupt task transitions, which is essentially latent forgetting.

Latent forgetting differs from catastrophic forgetting in CL Shoham et al. (2019); Xu et al. (2022); Durmus et al. (2021); Babakniya et al. (2023), where tasks are predefined and old classes are replaced by new ones at known boundaries. In FL, latent forgetting emerges gradually due to uneven, client-dependent data availability over time, which biases local gradients away from under-represented regions of data distribution and causes the global model to drift without the introduction of entirely new tasks or classes. Latent forgetting can be of two types: short-term, caused by temporary drops in data, and long-term, where under-representation persists over many rounds leading to lasting performance degradation.

## 3.2 FEDMEMO

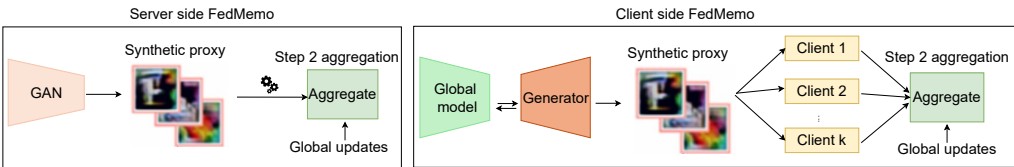

Figure 1: Overview of FedMemo with server-side approach and client-side approach.

We now present an overview of our proposed framework, FedMemo, a proxy-based 2-step aggregation method to mitigate latent forgetting in FL, applicable on both server and client sides (Figure 1). Each round begins with standard FL training and aggregation, which we call *step-1*. The server then computes weight update variance (from the aggregated updates) to monitor latent forgetting. If a decreasing trend is detected, either *server-side* or *client-side* FedMemo is triggered (see details in Section 4.1 and Appendix B). In *server-side* mode, the server trains the global model on synthetic proxy data (generated via GAN) and combines the resulting update with step-1 weights through another aggregation step, which we refer to as *step-2* aggregation. The final model after this step is shared with clients. In *client-side* mode, the server sends a generator to clients, to generate synthetic data and train it. Clients return updates, which the server again integrates via step-2 aggregation. The reasoning behind this 2-step process is that at *step-1 aggregation*, the global model is specialized in the real data and does not learn the decision boundary for the missing data. In order to prevent the model reducing ability to generalize representations for the missing data, the server applies the *step-2 aggregation* to trade off between learning the remaining and missing data. The final weights is formulated as:

$$w_{2step}^{t+1} = \frac{\sum_{i \in K_t} N_i w_i^{t+1} + N_p w_p^{t+1}}{\sum_{i \in K_t} N_i + N_p}, \quad w_p^{t+1} = w^{t+1} - \eta G_p(w^{t+1}), \tag{1}$$

where $G_p(w^{t+1}) := \mathbb{E}_{(x,y) \sim P_p}[\nabla_w \ell(f(x; w^{t+1}, y)]$, $P_p$ is the distribution, $N_p$ is the proxy data size.

Prior work on synthetic data in FCL Arjovsky et al. (2017); Rasouli et al. (2020); Babakniya et al. (2023) blindly mixes generated and real data at the client side without adapting to distribution shifts or knowing when to generate and use proxy data. In contrast, FedMemo separates proxy training into a second step which offers two key advantages: (1) it allows the model to revisit forgotten knowledge without interfering with ongoing local learning, reducing conflicting gradient directions and improving training stability; and (2) it enables proxy training on the server side, reducing client resource constraints. Proxy data in FedMemo is not blindly used, clients receive the generative model only when the server detects distribution shifts, as described in Section 3.4. We discuss the limitations of our framework in Appendix H.

## 3.3 THEORETICAL ANALYSIS

We first analyze how feature unavailability due to temporal-spatial heterogeneity causes performance degradation and latent forgetting in FL, and then show how the proposed FedMemo with the 2-step aggregation mitigates this issue. Consider a reference FL baseline without any heterogeneity. For a set of clients $K_t \subseteq K$ participating in round $t$, let $N_i^{ref}$ denote the size of data in client $i$ in the baseline and $N_i^t$ size of it after data encounters non-stationary availability . $P_i^{ref}(x, y)$ and $P_i^t(x, y)$ are the distribution for client $i$ for the baseline and the feature unavailability. We define $\alpha_i^{ref} = \frac{N_i^{ref}}{\sum_{i \in K_t} N_i^{ref}}$, $\alpha_i = \frac{N_i^t}{\sum_{i \in K_t} N_i^t}$, and $\Delta \alpha_i = \alpha_i^{ref} - \alpha_i^t$ the change in weights due to unavailable features in data distribution. Let $G_i^{ref}(w^t)$ and $G_i(w^t)$ denote local gradients under the reference and unavailable feature distributions, and let $G^{ref}(w^t)$ and $G(w^t)$ be the aggregated global gradients. We assume that round $t$ is the first round where the temporal-spatial heterogeneity bias appears and compare the reference loss under the reference update $w_{ref}^{t+1}$ and update $w^{t+1}$ for unavailable features case.

**Proposition 3.1** *Let $w_{ref}^{t+1}, w^{t+1}$ denote the reference update and model update with unavailable feature. where $G_i^{ref}(w^t), G_i(w), G^{ref}(w^t), G(w^t)$ denote the client and global gradients. Then,*

$$\begin{aligned} \mathcal{L}^{ref}(w^{t+1}) - \mathcal{L}^{ref}(w_{ref}^{t+1}) &= \eta \sum_{i \in K_t} \alpha_i^{ref} \langle G^{ref}(w^t), \Delta G_i(w^t) \rangle \\ &\quad + \eta \sum_{i \in K_t} \Delta \alpha_i \langle G^{ref}(w^t), G_i(w^t) \rangle + R^{(2)} \end{aligned}$$

*where $\Delta G_i(w^t) := G_i^{ref}(w^t) - G_i(w^t), \Delta\alpha_i := \alpha_i^{ref} - \alpha_i$, and $R^{(2)} = O(\eta^2)$*

Detailed proof of Proposition 3.1 can be found in Appendix C.1. The first term represents class deficit (the effect of under-represented classes), and the second term corresponds to the reweighting bias arising from shifts in client weights due to changing data quantities, i.e. temporal heterogeneity. The second-order remainder and can be neglected since the learning rate is usually small and $O(\eta^2)$ decays fast. We can bound the reweighting bias using the triangle inequality and Cauchy–Schwarz by $\left\|G^{ref}(w^t)\right\| \cdot \sum_{i \in K_t} |\Delta\alpha_i| \cdot \|G_i(w^t)\|$. Even if the client weights change only slightly due to feature unavailability causing temporal-spatial heterogeneity $|\Delta\alpha_i|$ are small, the reweighting bias can be a small correction term, But the first term is still dominated by class deficits. For under-represented classes, $\Delta G_i(w^t)$ measures the reduction in gradient contribution due to missing data in client $i$. $G_i(w^t)$ is smaller than $G_i^{ref}(w^t)$ because $G_i(w^t)$ under-represents those class gradients, therefore $\Delta G_i(w^t)$ which captures the deficits of those missing gradients aligns positively with $G^{ref}$ and reinforces the magnitude of the loss gap. However, for the well represented classes, $\Delta G_i(w^t)$ is small as they are close to the reference gradients and even if negative, it's smaller compared to the class deficits. So, from proposition 3.1 we have the following remark:

**Remark 3.1** *The reference loss evaluated at the weights for the model with unavailable features increases relative to the reference-update weights and causes latent forgetting.*

Next we show how 2-step aggregation mitigates this latent forgetting.

**Proposition 3.2** *let $w_{2step}^{t+1}$, $G_p(w^{t+1})$ the 2-step updates and updates for the unavailable features model where $G_p(w^{t+1})$ is the proxy gradient and $\beta = \frac{N_p}{\sum_{i \in K_t} N_i + N_p}$, Then,*

$$\mathcal{L}^{ref}(w_{2step}^{t+1}) - \mathcal{L}^{ref}(w^{t+1}) = -\eta(1 - \beta)\langle G^{ref}(w^{t+1}), G_p(w^{t+1})\rangle + R^{(2)} \qquad (2)$$

*where $R^{(2)} = O(\eta^2(1 - \beta)^2)$.*

Detailed proof of Proposition 3.2can be found in Appendix C.2. The first-order term is negative if $G_p(w^{t+1})$ aligns with $G^{ref}(w^{t+1})$ since $-\eta(1 - \beta) < 0$ and then 2-step aggregation reduces the reference loss. The remainder $O(\eta^2(1 - \beta)^2)$ is relatively small and negligible. According to the cosine similarity,

$$\cos(\theta) = \frac{\langle G_p(w^{t+1}), G^{ref}(w^{t+1})\rangle}{\|G_p(w^{t+1})\| \, \|G^{ref}(w^{t+1})\|}$$

where $cos(\theta)$ determines the quality of the proxy data and when $cos(\theta) > 0$, the proxy gradient helps mitigate forgetting.

### 3.4 FL LATENT FORGETTING DETECTION

Spatial heterogeneity in FL is typically coarse-grained and static, however, temporal-spatial heterogeneity is fine-grained and dynamic, as a result of evolving data distributions—e.g., features or classes becoming under-represented or unavailable over time. This temporal shift makes it significantly harder to detect than spatial heterogeneity alone. Most state-of-the-art methods detect only static spatial heterogeneity, assuming data distributions remain fixed throughout training. For instance, Yuan et al. (2022) models spatial variation but ignores temporal shifts. While MFCL Babakniya et al. (2023) and Yu et al. (2025) in FCL considers distributional changes, they assumes synchronized, fixed shifts across clients at sequential task boundaries. In reality, FL often involves asynchronous, uncoordinated feature shifts that are overlooked by these assumptions. Therefore, existing methods for detecting temporal-spatial heterogeneity are inadequate. Next, we propose a novel metric that identifies latent forgetting resulting from this temporal-spatial heterogeneity.

**Weight update variance:** Let $\Delta w^{(t)} = (w^{t+1}) - (w^t) = -\eta\nabla L(w^t)$ represent the weight update where $\Delta w^{(t)} = [\Delta w_1^{(t)}, \Delta w_2^{(t)}, \dots, \Delta w_d^{(t)}]$ and $d$ is the total number of parameters, then the weight update variance is:

$$WV^{(t)} = \tfrac{1}{d}\sum_{j=1}^{d}(\Delta w_j^{(t)} - \mu^{(t)})^2, \quad \mu^{(t)} = \tfrac{1}{d}\sum_{j=1}^{d}\Delta w_j^{(t)}$$

When short-term or long-term latent forgetting occurs, clients train on heterogeneous, limited data. Due to temporal-spatial heterogeneity, gradients from missing classes diminish, causing local updates

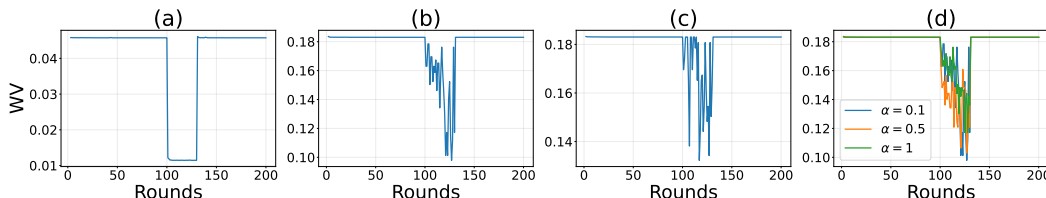

Figure 2: Weight update variance for (a) full missing data of 5 classes, (b) gradual and partial missing data for 3 clients, (c)gradual and partial feature unavailability for only 1 client, and (d) gradual and partial feature unavailability with different Non-IID levels - CIFAR10.

to shrink in those directions. Consequently, global updates focus more on well-represented classes, steering the model away from unavailable features of under-represented classes and resulting in poor generalization across the full data distribution.

We illustrate this with empirical observations. Figure 2(a) shows weight update variance on CIFAR-10 with removing of 5 classes between rounds 100–130 (removing their features entirely from the training pool). Figures 2(b,c) depict gradual unavailability of $30\%$ to $90\%$ in two classes (1 and 5) on 3 and 1 clients. Figure 2(d) shows how the detection metric behaves under varying Non-IID levels. As clients train on limited data, their updates align more on the limited features, reducing weight update variance across rounds. This drop signals latent forgetting, reflecting low update diversity using only recent server values. Additional analyses of client and proxy gradient variance and cosine similarities are in Appendix G. As a result, weight update variance captures latent forgetting in FL by reflecting the gradual loss of under-represented data impact in real time. Our approach is privacy-preserving and does not require sharing local weights. It also complies with differential privacy constraints, as detailed later in this Section.

**Proxy dataset generation:** In practical FL scenarios, proxy datasets are commonly used to design model architectures and tune global hyperparameters. These datasets are typically sourced from public repositories or collected with user consent, and are often provided by model developers during the initial setup and configuration phases. But this proxy data are usually small in size, so we use it to train the GAN in server. Following Zawad et al. (2025) the training data for generator contains 5000 samples and are excluded from the training set for FL. For the client side method, a similar generative model to the one in the Qi et al. (2023) is used in the server. Note that our client side method is different than MFCL, as FedMemo does not mix the generated synthetic data to the local real data and use a 2-step aggregation by training the real and synthetic data separately. For the GLUE dataset, we use the method from Meng et al. (2022), a zero-shot data generation approach. Additional details on the data generation process are provided in Appendix D.

Table 1: Summary of experimental baselines and proposed methods

| Method | Description |
|---|---|
| **No feature unavailability (baseline)** | No changes in data and no forgetting - FedAvg |
| **No feature unavailability + proxy (baseline)** | No changes in data and trained with added proxy data |
| **FL aggregation methods** | SCAFFOLD, FedProx |
| **SOTA forgetting prevention in FCL** | MFCL (only applicable for vision datasets) |
| **Client side FedMemo (ours)** | clients' generated synthetic data with 2-step aggregation. |
| **Server side FedMemo (ours)** | GAN-generated data in server with 2-step aggregation. |

**Privacy:** We specially emphasize on not using any private information about the clients including data distribution, number of data, number of classes, etc at any time during training. To improve the privacy, FedMemo can be implemented using privacy protection methods such as differential privacy (DP) Wei et al. (2020) to defend against attacks that attempt to extract private information about clients. Note that FedMemo is using aggregated weights and can be combined with DP. In FedMemo first aggregation step, each client uses a local $(\epsilon, \delta)$-DP algorithm where $\epsilon$ bounds the influence each client may have on the algorithm's output and $\delta$ defines the probability that this limit is exceeded. With random client selection (rate $q = \frac{|k_t|}{|K|}$), the privacy of the round would be $(O(q\epsilon), q\delta)$ as standard FL. In the second step, server updates the model using synthetic proxy data $D_p$ and is initialized from $w_{t+1}$. This step is independent of clients data and does not require additional privacy-preserving technique, since it does not impact the client level privacy (if synthetic data is on

client side, similar privacy as step 1 is applied). Final model $w'_{t+1}$ includes a differentially private aggregation over clients data and updates from synthetic proxy data and will remain $(O(q\epsilon), q\delta)$-DP.

## 4 EVALUATIONS

**Datasets, models, and hyperparameters -** We evaluate our method on CIFAR-10 Krizhevsky et al. (2009), SVHN Netzer et al. (2011), and GLUE tasks (CoLA, SST-2, MRPC, QQP, MNLI, QNLI, RTE) Wang et al. (2018). For CIFAR-10 and SVHN, we use ResNet-18 He et al. (2016) with 50 clients, 5 sampled per round, over 200 rounds. For GLUE, we use T5-large Colin (2020) with multi-task prefix tuning, training over 50 rounds with 35% of 467 clients sampled per round. We freeze all model parameters and apply prefix tuning to the final layer and for the detection on GLUE, we use only the weight gradient variance of the final layer. For proxy data generation in client-side we use generative model from Qi et al. (2023) and server-side GANs (vision) or zero-shot generation (GLUE) Meng et al. (2022). Full training and model details are provided in Appendix E.

**Data Distribution and latent forgetting setup -** Following Reddi et al. (2020), we use Dirichlet data distributions with $\alpha = 0.1$, where smaller $\alpha$ values correspond to higher levels of non-IID data. While we also evaluate more balanced distributions ($\alpha = 1$), where our method performs similarly with or slightly better than state-of-the-art approaches, we focus on heterogeneous settings where data shifts over time leads to more significant forgetting. Results for balanced setting are provided in Appendix F. To study latent forgetting, we simulate temporal-spatial heterogeneity through data drop patterns within a standard FL setup. For vision tasks, we gradually remove 30–90% of two classes (1 and 5) (details in Appendix D) for 3 clients per round, thus removing their features from the training pool. For GLUE tasks, we remove 30–70% features of 1 class in 40% of the clients. We examine both long-term (persistent data drops from rounds 100 for vision and 25–50 for GLUE) and short-term (temporary missing of features between rounds 100–130 for vision and 25–34 for GLUE) latent forgetting. Latent forgetting is detected based on weight update variance, and our 2-step aggregation is only triggered when such forgetting is detected, minimizing overhead. We also analyze long-term latent forgetting at different training stages: when the model has mostly converged (round 100), or during critical learning periods (early rounds such as 6).

To explore how synthetic data generation could help mitigate forgetting in FL, we adapt the state-of-the-art MFCL, We consider MFCL a representative and stronger baseline in replay for FCL. MFCL trains a generative model to synthesize data for forgotten tasks and distill knowledge back to clients. We adapt this approach in standard FL with simulating task shifts by missing features of some classes (5 for vision and 1 for GLUE) to serve as a naïve synthetic replay baseline for temporal-spatialheterogeneity (MFCL is only applicable for vision tasks). All baselines are summarized in Table 1. We use 3 seeds for each experiment (we report the standard deviation across the seeds for GLUE tasks in the Appendix A due to space limitations)

Table 2: Performance (% accuracy) of different baselines on CIFAR-10 and SVHN datasets under long-term latent forgetting near convergence and during critical learning period.

| Method | Near convergence | | Critical learning period | |
|---|---|---|---|---|
| | CIFAR-10 | SVHN | CIFAR-10 | SVHN |
| (Baseline) No feature un-availability | $74.69 \pm 0.32$ | $90.15 \pm 0.43$ | $74.69 \pm 0.32$ | $90.15 \pm 0.43$ |
| (Baseline) No feature un-availability + proxy | $79.54 \pm 0.13$ | $93.28 \pm 0.21$ | $79.54 \pm 0.13$ | $93.28 \pm 0.21$ |
| SCAFFOLD | $42.41 \pm 0.22$ | $55.03 \pm 0.12$ | $38.22 \pm 0.23$ | $53.91 \pm 0.18$ |
| FedProx | $42.58 \pm 0.43$ | $52.32 \pm 0.41$ | $37.15 \pm 0.16$ | $54.12 \pm 0.25$ |
| MFCL | $65.48 \pm 0.14$ | $88.57 \pm 0.11$ | $23.74 \pm 0.24$ | $43.58 \pm 0.31$ |
| Client sideFedMemo | $\mathbf{72.97 \pm 0.15}$ | $\mathbf{91.02 \pm 0.18}$ | $26.7 \pm 0.3$ | $52.19 \pm 0.21$ |
| Server-side FedMemo | $59.6 \pm 0.22$ | $67.09 \pm 0.25$ | $\mathbf{43.80 \pm 0.24}$ | $\mathbf{59.55 \pm 0.4}$ |

### 4.1 RESULTS

**Long-term latent forgetting near convergence -** When feature unavailability occurs at near conver-gence (round 100), results in Table 2 show that SCAFFOLD and FedProx suffer significant latent forgetting. Although, MFCL mitigates this to some extent, it still underperforms our approach. The client-side FedMemo achieves the best accuracy, outperforming MFCL by 7.49% on CIFAR-10 and 2.45% on SVHN. The server-side variant also improves over aggregation baselines, but client-side generation is more effective here, since the well-trained model enables the generator to synthesize high-quality, feature-representative data.

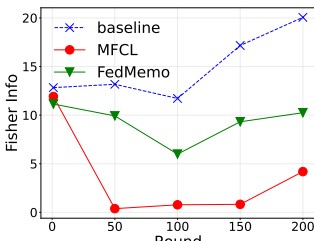

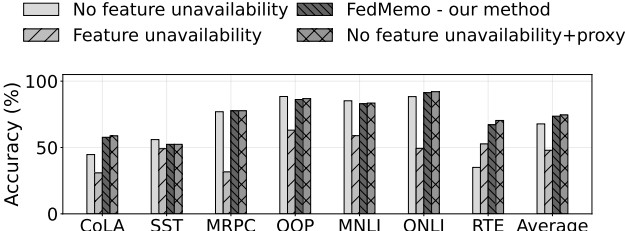

Figure 3: CIFAR10 dataset, Fisher information trace.

Figure 4: Accuracy of GLUE dataset after seasonal missing data

Table 3: Performance (% accuracy) of different baselines for CIFAR-10 and SVHN datasets in short-term latent forgetting; Columns 2,3: accuracy at the end of temporal missing data, columns 4,5: final accuracy

| Method | CIFAR-10 temporal | CIFAR10 final | SVHN temporal | SVHN final |
|---|---|---|---|---|
| (baseline) No feature unavailability | $70.36 \pm 0.21$ | $74.69 \pm 0.32$ | $87.80 \pm 0.12$ | $90.15 \pm 0.43$ |
| No feature unavailability + proxy | $74.31 \pm 0.11$ | $79.54 \pm 0.13$ | $91.02 \pm 0.22$ | $93.28 \pm 0.21$ |
| SCAFFOLD | $40.99 \pm 0.12$ | $73.52 \pm 0.32$ | $56.65 \pm 0.21$ | $88.34 \pm 0.3$ |
| FedProx | $39.25 \pm 0.22$ | $71.42 \pm 0.17$ | $55.26 \pm 0.18$ | $88.65 \pm 0.26$ |
| MFCL | $65.03 \pm 0.22$ | $71.12 \pm 0.23$ | $84.50 \pm 0.2$ | $91.06 \pm 0.23$ |
| Client side FedMemo | $\mathbf{72.07 \pm 0.21}$ | $75.54 \pm 0.13$ | $\mathbf{90.96 \pm 0.15}$ | $\mathbf{91.54 \pm 0.21}$ |
| Server side FedMemo | $60.37 \pm 0.12$ | $\mathbf{78.5 \pm 0.24}$ | $72.27 \pm 0.11$ | $89.87 \pm 0.23$ |

**Long-term latent forgetting during critical learning periods -** When features become permanently unavailable early in training, the client-side generator produces noisy synthetic samples, which degrade performance. This aligns with critical learning period findings Yan et al. (2021), where unrecovered early missing data leads to permanent accuracy loss. As shown in Table 2, MFCL fails to prevent latent forgetting and underperforms SCAFFOLD and FedProx, with the lowest accuracy on both datasets. Its low Fisher information trace (Figure 3) further confirms this. In contrast, server-side FedMemo achieves the best results. This reinforces that server-side proxy data is more reliable when latent forgetting occurs early, as client-side proxy data is too poor to prevent it effectively.

**Short-term latent forgetting -** Table 3 presents results for temporal unavailable features due to temporal-spatialheterogeneity, where we report both final accuracy and at the end of missing time. Client-side FedMemo achieves the highest accuracy during the feature unavailability phase, outperforming MFCL by $7.04\%$ and $6.46\%$ for CIFAR-10 and SVHN. Although server-side 2-step achieves higher final accuracy, it is less effective than the client-side approach during the dropped data interval, however, still outperforms state-of-the-art aggregation methods overall.

Table 4: GLUE dataset; task-wise and average % accuracy in multi-task learning, including gradual feature unavailability scenario

| Method | CoLA | SST2 | MRPC | QQP | MNLI | QNLI | RTE | Avg. |
|---|---|---|---|---|---|---|---|---|
| (baseline) No feature unavailability | 44.67 | 55.96 | 76.96 | 88.45 | 85.21 | 88.35 | 35.01 | 67.80 |
| No feature unavailability + proxy | 58.77 | 52.41 | 77.71 | 86.84 | 83.51 | 92.05 | 70.39 | 74.52 |
| FedProx | 30.87 | 49.08 | 31.61 | 63.08 | 58.90 | 49.36 | 52.70 | 47.94 |
| Server side FedMemo | **57.71** | **52.40** | **77.69** | **86.17** | **82.96** | **91.30** | **67.14** | **73.63** |
| Gradual feature unavailability | | | | | | | | |
| FedProx | 64.62 | 50.34 | 38.72 | 88.29 | 25.34 | 10.01 | 37.18 | 44.92 |
| Server side FedMemo | **52.92** | **52.06** | **77.45** | **86.26** | 83.45 | **91.21** | **70.75** | **73.44** |

Table 5: Gradual feature unavailability

| Method | CIFAR-10 short-term | CIFAR10 long-term | SVHN short-term | SVHN long-term |
|---|---|---|---|---|
| (baseline) No feature unavailability | $74.69 \pm 0.32$ | $74.69 \pm 0.32$ | $90.15 \pm 0.43$ | $90.15 \pm 0.43$ |
| SCAFFOLD | $69.43 \pm 0.22$ | $68.44 \pm 0.17$ | $84.57 \pm 0.22$ | $81.25 \pm 0.24$ |
| FedProx | $68.83 \pm 0.32$ | $67.23 \pm 0.31$ | $84.81 \pm 0.24$ | $80.93 \pm 0.27$ |
| Client side 2-step FedMemo | $\mathbf{74.51 \pm 0.15}$ | $\mathbf{74.07 \pm 0.12}$ | $\mathbf{90.96 \pm 0.2}$ | $\mathbf{90.72 \pm 0.19}$ |
| Server side 2-step FedMemo | $74.13 \pm 0.18$ | $72.2 \pm 0.14$ | $89.42 \pm 0.21$ | $88.41 \pm 0.2$ |

**Gradual feature unavailability -** Table 5 is the comparison of our method with state-of-the-art aggregation methods and shows that our method prevents the short-term and long-term latent forgetting in a normal FL with partial gradual data shifts causing temporal-spatialheterogeneity.

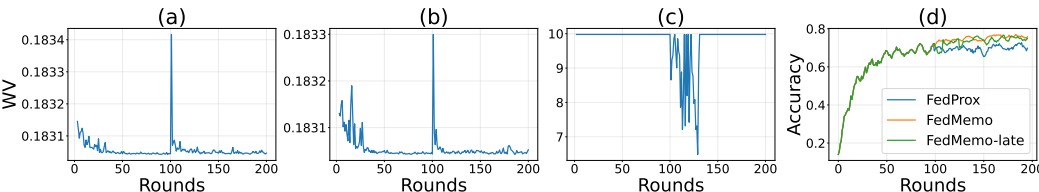

Figure 5: Robustness of FedMemo detection method (a) varying optimizer, (b) varying learning rate decay, (c) Detection using only last 5 layers weight updates, (d) FedMemo accuracy on CIFAR10 in standard FL gradual missing with a 8 rounds delay in applying FedMemo .

**Language classification tasks -** For GLUE, MFCL is not applicable, so we compare FedMemo against the strongest baseline aggregation methods (Appendix A contains full long-term latent forgetting results). Table 4 show results for temporal feature unavailability in multi-task prefix tuning. FedMemo improves over FedProx by $25.69\%$ in the temporal feature unavailability period while maintaining performance close to the baseline with proxy data (Figure 4). Notably, CoLA and RTE converge early due to their small size, but training continued for fairness across tasks. To our knowledge, this is the first work to address latent forgetting in multi-task prefix tuning for LLMs.

**Language classification tasks (gradual data loss) -** The results, summarized in Table 4, show that FedMemo significantly mitigates latent forgetting under this realistic data unavailability. Notably, it achieves a $16.4\%$ higher average accuracy during the missing-data period compared to the best baseline (FedProx), while maintaining performance close to the ideal baseline with proxy data. This demonstrates FedMemo's robustness in maintaining performance under cross-client label imbalance.

**FedMemo Robustness -** We evaluate the robustness of our detection method under optimizer and learning rate changes. As shown in Figure 5(a,b), switching from SGD to Adam or applying accelerated learning rate decay (after round 100) causes only short-term $WV$ spikes, whereas latent forgetting results in a sustained downward trend—allowing clear distinction between transient noise and true forgetting (CIFAR-10). We also test delayed detection: even when triggered 8 rounds late (at round 108), FedMemo still mitigates forgetting (Fig. 5(d)). Additionally, Figure 5(c) shows that computing $WV$ over only the last 5 layers is sufficient for accurate detection, making it a fast and reliable metric to use in large models.

Table 6: Varying proxy data quantity

| FedMemo | CIFAR-10 (%) |
| --- | --- |
| Server side (100%) | 59.6 |
| Server side (60%) | 57.94 |
| Server side (20%) | 55 |

Table 7: Varying proxy data quality

| FedMemo | CIFAR-10 (%) |
| --- | --- |
| Server side (100 rounds) | 59.6 |
| Server side (70 rounds) | 57.35 |
| Server side (50 rounds) | 54.93 |
| Server side (20% noise) | 58.14 |

**Sensitivity Analysis -** We evaluate the impact of proxy data quality and quantity on performance. As shown in Table 6, using over $60\%$ of the full proxy data results in only a $1.66\%$ drop in accuracy, and even with $20\%$ samples, the drop remains moderate at $4.6\%$. This demonstrates that FedMemo is robust even with limited proxy data, as the step-2 aggregation corrects for missing class representations. To assess quality, we vary the number of generator training rounds (50 and 70) and inject $20\%$ label noise (Table 7). Performance remains stable despite the degraded proxy quality, showing the method's robustness. In practice, since the server has sufficient resources to produce high-quality synthetic data, our approach effectively boosts the global model via 2-step aggregation.

## 5 CONCLUSION

We propose FedMemo, a privacy-preserving FL method to address latent forgetting caused by long-term or temporal feature unavailability. It detects latent forgetting from global model updates and uses proxy data—generated via a server-side GAN or client-side generator to preserve the learned features. A novel 2-step aggregation integrates proxy knowledge into global model training. We formulate temporal-spatial heterogeneity and perform a theoretical analysis of FedMemo and through extensive experiments demonstrate that it improves accuracy by up to $20.06\%$ during critical learning periods, and $7.49\%/7.04\%$ in long-term/short-term latent forgetting on CIFAR-10 and SVHN and outperforming state-of-the-art in FL methods by $25.69\%$ on the GLUE benchmark.

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

# A LONG-TERM LATENT FORGETTING ON LANGUAGE MODELS

Table 8 includes the results for long-term latent forgetting on GLUE dataset, FedMemo significantly decreases latent forgetting.

Table 8: Performance of GLUE dataset; task-wise and average accuracy in multi-task learning (split for readability)

| Method | CoLA | SST2 | MRPC | Avg. |
|---|---|---|---|---|
| (baseline) No missing data | $44.67 \pm 0.91$ | $55.96 \pm 0.6$ | $76.96 \pm 1.2$ | 67.80 |
| FedProx | $30.87 \pm 1.3$ | $47.23 \pm 0.3$ | $29.31 \pm 1.2$ | 44.32 |
| Server side FedMemo | $48.89 \pm 1.2$ | $51.35 \pm 0.7$ | $76.45 \pm 0.6$ | **72.6** |

| Method | QQP | MNLI | QNLI | RTE |
|---|---|---|---|---|
| (baseline) No missing data | $88.45 \pm 1.1$ | $85.21 \pm 0.72$ | $88.35 \pm 0.4$ | $35.01 \pm 1.4$ |
| FedProx | $63.08 \pm 1.4$ | $53.44 \pm 1.8$ | $45.38 \pm 2.1$ | $46.47 \pm 1.1$ |
| Server side FedMemo | $86.03 \pm 0.2$ | $83.17 \pm 0.2$ | $90.67 \pm 0.4$ | $71.64 \pm 0.3$ |

The standard deviations for GLUE, table 4 are listed below:

Table 9: GLUE dataset; task-wise and average accuracy in multi-task learning, including gradual data missing scenario (split for readability)

| Method | CoLA | SST2 | MRPC | QQP | Avg. |
|---|---|---|---|---|---|
| No missing data | $44.67 \pm 0.91$ | $55.96 \pm 0.6$ | $76.96 \pm 1.2$ | $88.45 \pm 1.1$ | 67.80 |
| No missing data + proxy | $58.77 \pm 0.8$ | $52.4 \pm 0.6$ | $77.7 \pm 0.7$ | $86.84 \pm 0.9$ | 74.52 |
| FedProx | $30.87 \pm 1.2$ | $49.08 \pm 0.2$ | $31.61 \pm 0.9$ | $63.08 \pm 1.6$ | 47.94 |
| Server side FedMemo | $57.71 \pm 0.4$ | $52.40 \pm 0.4$ | $77.69 \pm 0.3$ | $86.17 \pm 0.8$ | **73.63** |
| Gradual data missing | | | | | |
| FedProx | $64.62 \pm 0.8$ | $50.34 \pm 0.4$ | $38.72 \pm 1.1$ | $88.29 \pm 0.6$ | 44.92 |
| Server side FedMemo | $52.92 \pm 0.2$ | $52.06 \pm 0.4$ | $77.45 \pm 0.3$ | $86.26 \pm 0.5$ | **73.44** |

| Method | MNLI | QNLI | RTE | Avg. |
|---|---|---|---|---|
| No missing data | $85.21 \pm 0.7$ | $88.35 \pm 0.43$ | $35.01 \pm 1.4$ | 67.80 |
| No missing data + proxy | $83.51 \pm 0.64$ | $92.05 \pm 1.4$ | $70.39 \pm 0.8$ | 74.52 |
| FedProx | $58.90 \pm 2.1$ | $49.36 \pm 1.2$ | $52.70 \pm 0.8$ | 47.94 |
| Server side FedMemo | $82.96 \pm 0.7$ | $91.30 \pm 0.2$ | $67.14 \pm 0.4$ | **73.63** |
| Gradual data missing | | | | |
| FedProx | $25.34 \pm 2.1$ | $10.01 \pm 2.4$ | $37.18 \pm 0.7$ | 44.92 |
| Server side FedMemo | $83.45 \pm 0.4$ | $91.21 \pm 0.2$ | $70.75 \pm 0.7$ | **73.44** |

# B ANOMALY DETECTION

**Statistical methods** We can use statistical anomaly detection methods to identify anomalies that significantly deviate from this expected pattern. One common statistical approach is moving average with threshold. This method detects anomaly points by comparing each datapoint to the average of a recent reference window. Once an anomaly is detected, we keep the reference window frozen until data returns to normal levels to prevent. In our experiments, we could detect all anomaly points using this method. Following figure shows anomalies detected by this method for CIFAR-10 dataset on temporal latent forgetting in rounds 100-130. We found the WV-based detector is robust to threshold choice across drop ratios from 0.3 to 0.7, precision, recall, and F1-score remain 0.96.

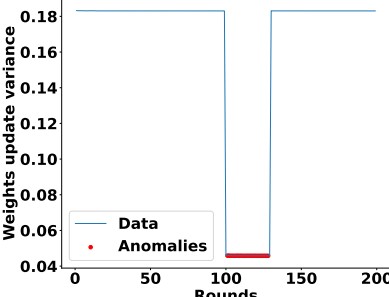

Figure 6: Anomalies for weight update variance for CIFAR-10

**Machine learning based methods** However, statistical methods were effective in our experiments, more sophisticated approaches may be better suited for real-time anomaly detection. To this end, we

employed the method proposed in Tuli et al. (2022) on weight update variance for CIFAR-10 dataset achieving a high F1-score of 98.2%, in detecting anomaly points.

## C.1 PROOF OF PROPOSITION 3.1

$w_{ref}^{t+1} = w^t - \eta G^{ref}(w^t), w^{t+1} = w^t - \eta G(w^t)$, are the reference model and model trained under missing data updates, $P_i^{ref}(x, y) = P_i^{ref}(y = c)P_i^{t,ref}(x|y = c)$, and and $P_i^t(x, y) = P_i^t(y = c)P_i^t(x|y = c)$ are the distribution for client $i$ at round t for the baseline and the missing case.

The local gradients are as following:
$$G_i^{\mathrm{ref}}(w^t) := \nabla\mathcal{L}_i^{\mathrm{ref}}(w^t) = \sum_{c=1}^{C} P_i^{\mathrm{ref}}(c)\, \mathbb{E}_{x\sim P_i^{\mathrm{ref}}(X|Y=c)}\left[\nabla_w \ell(f(x;w^t), c)\right],$$
$$G_i(w^t) = \nabla\mathcal{L}_i(w^t) = \sum_{c=1}^{C} P_i^t(c)\, \mathbb{E}_{x\sim P_i^t(X|Y=c)}\left[\nabla_w \ell(f(x;w^t), c)\right]$$
where
$$\mathcal{L}_i^{ref}(w^t) = \sum_{c=1}^{C} P_i^t(c)\mathbb{E}_{x\sim P_i^{ref}(X|Y=c)}\left[\ell(f(x;w^t), c)\right],$$
$$\mathcal{L}_i(w^t) = \sum_{c=1}^{C} P_i^t(c)\mathbb{E}_{x\sim P_i^t(X|Y=c)}\left[\ell(f(x;w^t), c)\right]$$

Then the global gradients and global loss are as following:
$$G^{\mathrm{ref}}(w^t) := \sum_{i\in K_t} \alpha_i^{\mathrm{ref}} G_i^{\mathrm{ref}}(w^t); \quad \mathcal{L}^{\mathrm{ref}}(w^t) = \sum_{i\in K_t} \alpha_i^{\mathrm{ref}} \mathcal{L}_i^{\mathrm{ref}}(w^t),$$
$$G(w^t) := \sum_{i\in K_t} \alpha_i G_i(w^t); \quad \mathcal{L}(w^t) = \sum_{i\in K_t} \alpha_i \mathcal{L}_i(w^t)$$

*proof.* The Taylor expansion of $L^{ref}(w^{t+1})$ and $L^{ref}(w^{t+1})$ at $w^t$ is as following:
$$
\begin{aligned}
\mathcal{L}^{ref}(w_{ref}^{t+1}) &= L^{ref}(w^t) + \langle\nabla L^{ref}(w^t), w_{ref}^{t+1} - w^t\rangle + \tfrac{1}{2}(w_{ref}^{t+1} - w^t)^\top \nabla^2 \mathcal{L}^{ref}(\zeta_a)(w_{ref}^{t+1} - w^{t+}) \\
&= L^{ref}(w^t) + \langle G^{ref}(w^t), -\eta G^{ref}(w^t)\rangle + \tfrac{1}{2}(w_{ref}^{t+1} - w^t)^\top \nabla^2 \mathcal{L}^{ref}(\zeta_a)(w_{ref}^{t+1} - w^{t+}) \\
&= L^{ref}(w^t) - \eta\left\|G^{ref}\right\|^2 + \tfrac{1}{2}(w_{ref}^{t+1} - w^t)^\top \nabla^2 \mathcal{L}^{ref}(\zeta_a)(w_{ref}^{t+1} - w^t)
\end{aligned}
$$
$$
\begin{aligned}
\mathcal{L}^{ref}(w^{t+1}) &= L^{ref}(w^t) + \langle\nabla L^{ref}(w^t), w^{t+1} - w^t\rangle + \tfrac{1}{2}(w^{t+1} - w^t)^\top \nabla^2 \mathcal{L}^{ref}(\zeta_b)(w^{t+1} - w^t) \\
&= L^{ref}(w^t) + \langle G^{ref}(w^t), -\eta G(w^t)\rangle + \tfrac{1}{2}(w^{t+1} - w^t)^\top \nabla^2 \mathcal{L}^{ref}(\zeta_b)(w^{t+1} - w^t) \\
&= L^{ref}(w^t) - \eta\langle G^{ref}, G(w^t)\rangle + \tfrac{1}{2}(w^{t+1} - w^t)^\top \nabla^2 \mathcal{L}^{ref}(\zeta_b)(w^{t+1} - w^t)
\end{aligned}
$$
Then:
$$
\begin{aligned}
\mathcal{L}^{ref}(w^{t+1}) - \mathcal{L}^{ref}(w_{ref}^{t+1}) &= \eta(\left\|G^{ref}(w^t)\right\|^2 - \langle G^{ref}(w^t), G(w^t)\rangle) + R^2 \\
&= \eta\langle G^{ref}(w^t), G^{ref}(w^t) - G(w^t)\rangle + R^2 \\
&\overset{1}{=} \eta\sum_{i\in K_t} \alpha_i^{\mathrm{ref}}\langle G^{\mathrm{ref}}(w^t), \Delta G_i(w^t)\rangle + \eta\sum_{i\in K_t} \Delta\alpha_i\langle G^{\mathrm{ref}}(w^t), G_i(w^t)\rangle + R^2
\end{aligned}
$$
Where

$$R^2 = \frac{1}{2}(w^{t+1} - w_{ref}^{t+1})^\top \nabla^2 \mathcal{L}^{ref}(\zeta_1)(w^{t+1} - w_{ref}^{t+1}) = O(\eta^2)$$

for some $\zeta_1$ on the line segment between $w_{ref}^{t+1}$ and $w^{t+1}$ and equity 1 holds because:
$$
\begin{aligned}
G^{ref}(w^t) - G(w^t) &= \sum_{i\in K_t} \alpha_i^{ref} G_i^{ref}(w^t) - \sum_{i\in K_t} \alpha_i G_i(w^t) \\
&= \sum_{i\in K_t} \alpha_i^{ref}(G_i^{ref}(w^t) - G_i(w^t)) + \sum_{i\in K_t}(\alpha_i^{ref} - \alpha_i)G_i(w^t) \\
&= \sum_{i\in K_t} \alpha_i^{ref}(\Delta G_i(w^t)) + \sum_{i\in K_t} \Delta\alpha_i G_i(w^t)
\end{aligned}
$$

## C.2 PROOF OF PROPOSITION 3.2

*proof.* Let $w_{2\text{step}}^{t+1} = \beta w^{t+1} + (1 - \beta)w_p^{t+1}$, $\quad w_p^{t+1} = w^{t+1} - \eta G_p(w^{t+1})$ $\quad G_p(w^{t+1}) := \mathbb{E}_{(x,y)\sim P_p}[\nabla_w \ell(f(x;w^{t+1}, y)]$, and $D_p$ is the proxy data. where $P_p$ is the distribution induced by proxy data $D_p$. Then the second-order Taylor expansion of $L^{ref}(w_{2step}^{t+1})$ at $w^{t+1}$ is:
$$
\begin{aligned}
L^{ref}(w^{t+1}) &= L^{ref}(w^{t+1}) + \langle\nabla L^{ref}(w^{t+1}), w_{2step}^{t+1} - w^{t+1}\rangle + R^2 \\
&= L^{ref}(w^{t+1}) + \langle G^{ref}(w^{t+1}), (1 - \beta)(w_p^{t+1} - w^{t+1})\rangle + R^2 \\
&= L^{ref}(w^{t+1}) + (1 - \beta)\langle G^{ref}(w^{t+1}), (w^{t+1} - \eta G_p(w^{t+1}) - w^t + \eta G(w^t))\rangle + R^2 \\
&= L^{ref}(w^{t+1}) + (1 - \beta)\langle G^{ref}(w^{t+1}), (w^{t+1} - \eta G_p(w^{t+1}) - w^t + \eta G(w^t))\rangle + R^2 \\
&= L^{ref}(w^{t+1}) + (1 - \beta)\langle G^{ref}(w^{t+1}), (-\eta G(w^t) - \eta G_p(w^{t+1}) + \eta G(w^t))\rangle + R^2 \\
&= L^{ref}(w^{t+1}) - \eta(1 - \beta)\langle G^{ref}(w^{t+1}), G_p(w^{t+1})\rangle + R^2
\end{aligned}
$$

where

$$R^2 = \frac{1}{2}\eta^2(1-\beta)^2 G_p(w^{t+1})^T \nabla^2 \mathcal{L}^{ref}(\zeta_2) G_p(w^{t+1}) = O(\eta^2(1-\beta)^2)$$

for some $\zeta_2$ on the line segment between $w_{2step}^{t+1}$ and $w^{t+1}$.

## D  PROXY DATA GENERATION FOR GLUE DATASET

**Hyperparameters** CIFAR-10 and SVHN use 200 rounds, local 3 epochs, batch size of 64, learning rate of 0.01, with 5 clients selected per round from 50 total clients. GLUE tasks use batch size 32, learning rate 0.01, 50 rounds, and $35\%$ of 467 clients selected per round. To construct synthetic training data, we adopt the SuperGen framework Meng et al. (2022). Following their setup, we use a large autoregressive language model (CTRL) as the generator to produce class-conditioned samples via label-descriptive prompts, and a bidirectional model (COCO-LM Large) as the classifier. The generated outputs are filtered based on likelihood scores, and the retained synthetic examples are then used to fine-tune the classifier with label smoothing and temporal ensembling, exactly as described in Meng et al. (2022). For the gradual feature unavailability we remove the data starting at round 100 with $30\%$ feature unavailability and then increases by 2 percentage points in each round, until reaching $90\%$ by round 130.

## E  GAN ARCHITECTURE

For the vision tasks on server-side FedMemo, we use following GAN architecture in server:

| Generator | Discriminator |
|---|---|
| FC(100, 4×4×256) | Conv2d(3, 64, 4, 2, 1) |
| reshape(-, 256, 4, 4) | BatchNorm(64) |
| BatchNorm(256) | LeakyReLU(0.2) |
| ReLU | Conv2d(64, 128, 4, 2, 1) |
| ConvTranspose2d(256, 128, 4, 2, 1) | BatchNorm(128) |
| BatchNorm(128) | LeakyReLU(0.2) |
| ReLU | Conv2d(128, 256, 4, 2, 1) |
| ConvTranspose2d(128, 64, 4, 2, 1) | BatchNorm(256) |
| BatchNorm(64) | LeakyReLU(0.2) |
| ReLU | Conv2d(256, 1, 4, 1, 0) |
| ConvTranspose2d(64, 3, 4, 2, 1) | Sigmoid |
| Tanh | |

## F  FEDMEMO PERFORMANCE ON BALANCED DATA DISTRIBUTION

Table 10 shows that FedMemo achieves performance comparable to MFCL under balanced client-side data distributions ($\alpha = 1$). This confirms that while MFCL is effective in settings with minimal spatial heterogeneity, FedMemo remains competitive—even without fine-tuned client-specific configurations. However, the goal of this paper is not to optimize performance under ideal conditions, but to address realistic federated learning challenges where data is often non-IID, imbalanced, or subject to unpredictable shifts.

Importantly, MFCL relies on custom definitions of when data is missing and lacks a built-in forgetting detection mechanism, limiting its adaptability in dynamic environments. In contrast, FedMemo incorporates an automated latent forgetting detector and flexibly adapts through proxy-based correction, making it more robust to temporal and spatial heterogeneity.

Furthermore, in resource-constrained settings where client-side retraining is impractical, FedMemo supports a server-side correction mechanism, which allows adaptation without burdening clients. Improving this server-side component is left as future work, but even in its current form, FedMemo offers a practical and generalizable solution for real-world federated learning.

## G  OTHER METRICS

**Gradient variance -** To illustrate the gradient variance, we use a synthetic proxy data in server trained on the global model of clients at each round.

| Method | CIFAR-10 | SVHN |
|---|---|---|
| (Baseline) No missing data | $85.63 \pm 0.22$ | $94.07 \pm 0.25$ |
| Missing data - SCAFFOLD | $50.07 \pm 0.15$ | $57.37 \pm 0.32$ |
| Missing data - FedProx | $51.07 \pm 0.19$ | $56.43 \pm 0.56$ |
| Missing data with SOTA forgetting prevention in CL (MFCL) | $84.03 \pm 0.17$ | $94.01 \pm 0.09$ |
| Client side 2-step FedMemo | $84.56 \pm 0.15$ | $94.74 \pm 0.12$ |
| Server side 2-step FedMemo | $68.25 \pm 0.42$ | $88.63 \pm 0.22$ |

Table 10: Performance of different baselines for CIFAR-10 and SVHN datasets in long-term latent forgetting- $\alpha = 1$.

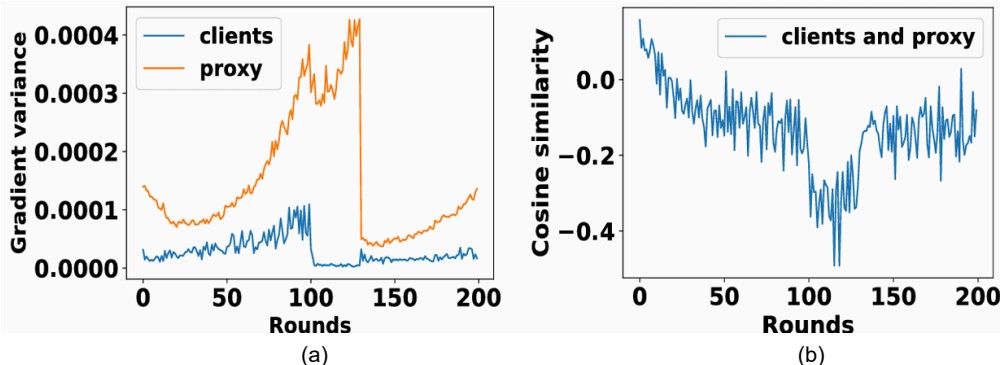

(a)             (b)

Figure 7: Synthetic proxy works as indicator of FL forgetting and be used as a support to prevent it after detection.

Let $g_{p,i}^{(t,l)} = \nabla L_{p,i}^{(T)}\left(\theta_i^{(l)}\right)$ represent the global (aggregated) gradient at round $t$ on layer $l$, and $k_p \subset P$ is set of participated samples of proxy $P$ at round t. Then, $g_p^{(t)} = [g_{p,i}^{(t,0)}, ..., g_{p,i}^{(t,L-1)}]$ denotes aggregated gradients at round $t$ over $l$ layers, and gradient variance is as follows:

$$GV^{(t)} = \frac{1}{d}\sum_{j=1}^{d}\left(G_{p,j}(w^{(t)}) - \mu^{(t)}\right)^2, \quad \text{where} \quad \mu^{(t)} = \frac{1}{d}\sum_{j=1}^{d}G_{p,j}(w^{(t)}) \tag{3}$$

where $d$ is the total number of parameters, and $G_{p,j}(w^{(t)})$ is the $j$-th coordinate of this vector.

**Cosine similarity of gradients -**

onsider $G_p(w^{(t)})$ as the synthetic proxy gradient at the server, and $G(w^{(t)})$ as the global gradient aggregated from clients. The cosine similarity is defined as:

$$\text{Cosine Similarity}(G(w^{(t)}), G_p(w^{(t)})) = \frac{G(w^{(t)}) \cdot G_p(w^{(t)})}{\|G(w^{(t)})\| \, \|G_p(w^{(t)})\|}. \tag{4}$$

We analyze two additional signals: clients and proxy data gradients variance in this section. equation 3 (Figure7 (a)). The clients' reduced gradient variance indicates that local clients gradients are becoming more homogeneous and previously learned information are forgotten. Note that, here we don't use proxy for mitigation and we are only using it as a reference (proxy is trained on server only and does not contribute to aggregation of local clients with missing data at all). The purpose is to observer the patterns resulted from missing data. In contrast, the synthetic proxy data preserves a more balanced view of the full distribution. As the global model forgets earlier patterns, the proxy gradients begin to diverge from client gradients and highlight the missing patterns. We observe this through a decline in cosine similarity (Figure7 (b)) between proxy and client gradients, which provides supporting evidence of the latent forgetting. While cosine similarity is not used as a detection signal, it helps interpret the shift in learning dynamics.

## H LIMITATIONS

For our client-side method, clients receive a shared generative model trained on server. Similar to the state-of-the-art method MFCL, clients need to generate proxy data which may introduce additional computation. However, unlike training, the computational overhead introduced by such generation

task is usually very small and this model is shared only once the data shifts is detected. In addition, we also offer a server-side solution that offloads such generation work to the server and thus impose no additional computation on

