# OpenReview forum: "Demystifying Latent Forgetting in Federated Learning"
_ICLR.cc/2026/Conference — Submitted to ICLR 2026_

### Official Review · Reviewer_zjdp · 2025-10-26

**Soundness:** 2
**Presentation:** 3
**Contribution:** 2
**Rating:** 2
**Confidence:** 3

**Summary:**

This paper aims to solve the spatial data heterogeneity in federated learning (FL), a feature-level latent forgetting.This paper proposes FedMemo, a privacy-preserving framework that employs an automated detection mechanism to detect latent forgetting in real-time. Extensive experiments show the benefits of FedMemo in language and vision tasks.

**Strengths:**

1. This work targets a less studied problem, temporal-spatial data heterogeneity, which is a critical problem in real-world applications.

2. While using the proxy data, the proposed method FedMemo ensures the privacy-preserving feature of FL, providing a secure way to use proxy data.This method also incurs minimal additional communication costs compared with previous approachs.

3. The experiments cover various tasks including vision and language tasks. The improved performance on these tasks show the effectiveness of the proposed method.

**Weaknesses:**

1. This work prooposes two methods, including client-side and server-side FedMemo. They have different benefits in critical learning period and near convergence, posting the problem when to use the client-side or server-side method. In practice, it is hard to identify which period the learning process is in.

2. The quality of proxy data has critical impact on the final model performance of FL. This implies the difficulty of training a GAN model in this setup. In practice, the proxy data is hard to obtain or has low quality or different distributions.

3. The experiments compare the method with three methods: SCAFFOLD and FedProx are not specially designed for the spatial heterogeneity problem, and MFCL is designed for vision tasks. The lack of baselines weakens the benefits of the proposed method.

**Questions:**

1. Appendix E provides the GAN architecture for the vision task, how about the language task?

2. If there are some proxy data, can we directly use other approachs without training GAN to improve the model performance? I believe these should be strong baselines for this work.

3. Using DP can enhance the privacy protection of FedMemo for sure, but could authors provide an analysis on the privacy leakage of FedMemo without DP?

---

> ### Author Response · Authors · 2025-11-24
> **Response to Official Review by Reviewer zjdp (1/2)**
>
> We thank reviewer zjdp for highlighting the importance of studying temporal–spatial data heterogeneity, and for recognizing both the privacy-preserving design and the strong empirical performance of FedMemo across vision and language tasks. We appreciate these encouraging comments and address the remaining concerns that help us improve our paper. Here is a brief summary:
>
> (1) We clearly explained when to use server-side versus client-side FedMemo and provided a direct comparison, while answering the ambiguity about critical learning period and its detection.
>
> (2) We clarified why a GAN is needed to amplify the small proxy dataset and why the raw proxy alone is insufficient.
>
> (3) We presented new results by adding comparisons with 2 more state-of-the-art methods, showing that FedMemo consistently outperforms them.
>
> (4) We clarified that FedMemo does not introduce additional privacy risk and is aligned with standard FL methods in privacy.
>
> (5) We explained how synthetic data is generated for NLP tasks following [1].
>
> Below, we provide the detailed responses for each comment.
>
> **1. When to Use Server-Side versus Client-Side FedMemo (Response to W2 and Q1:)**
>
> Below, we provide clear guidance and a concise comparison of the both variations.
>
> (a) Server-Side FedMemo:
>
> It can always be used and is especially recommended during early training (critical learning periods (CLP), and to detect CLP, one may rely on accuracy or use existing methods such as CriticalFL [2], which introduces a lightweight metric called the federated gradient norm (FGN)). Server-side FedMemo is also the best choice when clients have limited computation.
> | Pros| Cons
> |-|-|
> | -Zero additional client cost|- Focuses on global correction rather than local corrections|
> |-Reliable even when the global model is unstable|  |
> |-Privacy-preserving|  |
> |-Consistently improves forgetting during all training phases|  |
> |- No dependency on client hardware|  |
>
> (b) Client-Side FedMemo:
>
> When clients have sufficient compute resources and the global model has become stable—so the generator is well aligned with local distributions—the client-side variant can provide improvements over the server-side global correction, making client-side FedMemo advantageous.
>
> | Pros | Cons |
> |-|-|
> | - Adds client-specific correction | - Additional local computation (train on generated data)  |
> | - Yields the best final accuracy in near-convergence stages  | - Not the best during CLP due to weaker generator quality (a limitation shared by all client-side generative methods, including MFCL) |
> | - Privacy-preserving: clients generate synthetic data locally from a shared generator trained on server |  |
>
> One example of real world application is to use server-side and later switch to client-side if local resources are available. If local resources are limited, or if detecting CLP rounds / relying on accuracy is not desired, then server-side FedMemo would be used.
>
> **2. On the proxy data quality, and why not using it without training a GAN:**
>
> We thank the reviewer for the question. FedMemo does not require high-fidelity proxy samples, but it does require a proxy distribution that is sufficiently diverse to provide missing-feature gradient directions. The initial real proxy set is small and low-coverage; when trained centrally with ResNet18, it achieves only 34.26% on CIFAR-10 and 48.88% on SVHN, confirming that it cannot meaningfully approximate global structure or supply diverse gradients. This is why using the raw proxy directly—as suggested—is ineffective.
>
> The GAN is not introduced to achieve perfect generation, but to expand this tiny, balanced IID proxy into a broader synthetic surrogate distribution that covers missing-feature directions better than the raw proxy alone. Our sensitivity study (Table 7) shows that FedMemo remains stable even when the GAN is deliberately degraded (reduced training or label noise), demonstrating FedMemo’s robustness to low-quality synthetic generation.

---

> > ### Author Response · Authors · 2025-11-24
> > **Response to Official Review by Reviewer zjdp (2/2)**
> >
> > **3. New baselines:**
> >
> > We added 2 more competitive baselines from three representative families of methods:
> >
> > (1) regularization-based approaches for spatial heterogeneity (**FedBR**),
> >
> > (2) GAN-based aggregation (**FedMGD**), and
> >
> > (3) temporal–spatial heterogeneity methods from the FCL literature (**MFCL**).
> >
> > FedBR [3] is a strong method to reduce the learning bias on local features and classifiers. It outperforms FedNDT, MOON, and FedMix, and FedMGD [4] is a GAN-based aggregation method outperforming both Gan-based and Distillation-based methods such as FedDF, FedGEN, and FedGAN.
> >
> > FedMemo constantly outperforms all baselines. New results are in the following tables and FedMemo outperforms all baselines.
> >
> > **Short-term latent forgetting:**
> > | Method/ Dataset     | CIFAR10 (temporal) | CIFAR10 (final) | SVHN (temporal) | SVHN (final)
> > |-|-|-|-|-|
> > | FedBR    | 48.19    | 70.36      | 61.65  |  88.74 |
> > | FedMGD    | 49.14   | 71.23      |  66.1  |  89.77 |
> > |Client-side FedMemo|72.07 | 75.54| 90.96| 91.54|
> > |Server-side FedMemo| 60.37| 78.5| 72.27 | 89.87|
> >
> > **Long-term forgetting:**
> > | Method/ Dataset     | CIFAR10  | SVHN | CIFAR10 (CLP) | SVHN (CLP)
> > |-|-|-|-|-|
> > | FedBR    | 50.46    | 56.12      | 38.93  |  54.82 |
> > | FedMGD    | 51.74   | 58.57      |  39.54  |  55.24 |
> > |Client-side FedMemo|72.97| 91.02 | 26.7 | 52.19
> > |Server-side FedMemo| 59.6 | 67.09 | 43.80 | 59.55
> >
> > **Gradual feature unavailability:**
> > | Method/ Dataset     | CIFAR-10 (short-term) | CIFAR-10 (long-term) | SVHN (short-term) | SVHN (long-term)
> > |-|-|-|-|-|
> > | FedBR    | 70.02   | 68.73     | 85.23  |  83.20 |
> > | FedMGD    | 70.94   | 69.24      |  86.42  |  84.11 |
> > |Client-side FedMemo| 74.51| 74.07| 90.96 | 90.72|
> > |Server-side FedMemo| 74.13| 72.2 | 89.42| 88.41
> >
> > **4. FedMemo privacy:**
> >
> > FedMemo preserves the standard privacy guarantees of federated learning. Clients never send raw data, labels, or feature information:
> >
> > - In the server-side variant, all synthetic proxy data are generated and trained entirely on the server.
> >
> > - In the client-side variant, synthetic samples are created locally and never leave the device; clients transmit only their usual model updates from real-data and synthetic-data training.
> >
> > - The WV detector uses only the aggregated updates already present in any FL protocol.
> >
> > FedMemo introduces no new privacy risks beyond those already present in FL and is fully compatible with existing protections such as differential privacy or secure aggregation.
> >
> > **5. Synthetic Data Generation for NLP**
> >
> > For language tasks, we do not use a GAN. As noted in line 302, we follow [1], which generates synthetic text using a pretrained language model in a zero-shot manner. We used exactly the same setting as the paper and will add the details in Appendix. This method requires no task-specific data and therefore does not need any GAN specification. The zero-shot generator is not trained from the global FL model—it remains fixed and does not depend on client updates. As a result, all synthetic text generation can be performed entirely on the server, and because the generator is not client-dependent, a client-side FedMemo variant offers no additional benefit for NLP. We will clarify this in the revision.
> >
> >
> > [1] Yu Meng, Jiaxin Huang, Yu Zhang, and Jiawei Han. Generating training data with language models: Towards zero-shot language understanding. Advances in Neural Information Processing Systems, 35:462–477, 2022.
> >
> > [2]Yan, Gang, et al. "Criticalfl: A critical learning periods augmented client selection framework for efficient federated learning." Proceedings of the 29th ACM SIGKDD Conference on Knowledge Discovery and Data Mining. 2023.
> >
> > [3] Guo, Yongxin, Xiaoying Tang, and Tao Lin. "Fedbr: Improving federated learning on heterogeneous data via local learning bias reduction." International conference on machine learning. PMLR, 2023.
> >
> > [4] Sheng, Tao, et al. "Modeling global distribution for federated learning with label distribution skew." Pattern Recognition 143 (2023): 109724.

---

> ### Comment · Reviewer_zjdp · 2025-11-27
> **Thank you for response!**
>
> After reading the authors' responses, some of my concerns are addressed. I raise the score from 2 to 4.
> However, my concern about the proxy data remains, like Reviewer VDS8. In particular, how can we obtain a high-quality proxy data? If the proxy data has poor quality, how can we train GAN well? Despite the good results, I belive this assumption about proxy data is strong.

---

> > ### Author Response · Authors · 2025-11-28
> > **Addressing question about proxy and GAN**
> >
> > We sincerely thank the reviewer zjdp for further feedback and raising the score. We would like to respectfully clarify that FedMemo is explicitly designed to function without high-quality proxy data, nor does it require the proxy to approximate the full client data distribution.
> >
> > Unlike prior works that rely on high-fidelity public data for calibration, our method uses the proxy solely for coarse directional alignment of gradients. Below, we provide evidence and clarification on why FedMemo is robust to poor-quality proxies:
> >
> > 1. Empirical evidence of poor proxy quality in our evaluation. We validate that our raw proxy data is indeed of low quality. When training a ResNet-18 centrally on only the proxy data, it achieves just 34.26% accuracy on CIFAR-10 and 48.88% on SVHN. Despite this low quality proxy data used, FedMemo still achieves high performance, demonstrating that high-quality data is not a prerequisite.
> >
> > 2. Theoretical justification (coarse alignment). As detailed in our theoretical analysis (Section 3.3; Proposition 3.2), the Step-2 gradient correction only requires the proxy gradient to have a positive directional alignment (cosine similarity > 0) with the true gradient. It does not require the proxy samples to be high quality.
> >
> > 3. Robustness to GAN degradation. The GAN in FedMemo is not trained to replicate the full complexity of the client distribution. Its specific role is to expand the tiny proxy dataset into a broader synthetic surrogate that provides these coarse directional signals. Therefore, “training the GAN well” in our context implies only achieving this directional utility, which is much easier to satisfy than generating high quality samples. Our sensitivity study (Table 7) demonstrates that FedMemo’s performance remains stable even when the GAN is deliberately degraded (e.g., reducing training to 50–70 rounds or injecting 20% label noise). This confirms that the system does not require a highly optimized generator.
> >
> > In summary, the assumption that high-quality proxy data is needed is a misunderstanding of standard FL requirements which FedMemo overcomes. Our unique gradient correction mechanism allows us to bypass the need for high-fidelity proxies entirely (validated by both theoretical analysis and empirical evaluation).

---

### Official Review · Reviewer_P3Tj · 2025-10-30

**Soundness:** 2
**Presentation:** 2
**Contribution:** 2
**Rating:** 4
**Confidence:** 3

**Summary:**

This paper proposes FedMemo, a privacy-preserving federated learning framework designed to detect and mitigate feature-level latent forgetting caused by temporal–spatial data heterogeneity. The method introduces an automated forgetting detection mechanism based on global weight updates and a proxy-based two-step aggregation strategy to alleviate performance degradation from missing features. This work introduces a new concept of “latent forgetting,” but there is still room for improvement in the aspects of method design, experimental design, and overall writing quality.

**Strengths:**

1. The paper presents a new perspective by introducing latent forgetting caused by feature unavailability, which differs from catastrophic forgetting in continual learning.
2. It provides a certain level of theoretical analysis to explain the phenomenon of latent forgetting.
3. The overall writing structure is clear and well organized, but some parts of the content are not clearly described.

**Weaknesses:**

1. The authors describe two types of latent forgetting, but the method section does not clearly explain how the proposed approach effectively addresses both. Evidence based solely on accuracy results in the experiments is insufficient to support this claim.
2. In FedMemo, latent forgetting is detected through Weight Update Variance. However, relying only on Weight Update Variance is inadequate, since its decrease can result from various factors such as convergence, and does not necessarily indicate latent forgetting.
3. Figure 1 illustrating FedMemo is not sufficiently clear, particularly regarding how the generative model operates differently on the server and client sides.
4. The proposed “2-step aggregation” appears to be a simple proxy-based fine-tuning procedure and lacks substantial innovation.
5. The paper conflates feature unavailability with sample unavailability. Specifically, most experiments simulate “feature unavailability” by removing certain classes, which in fact represents class or sample missingness, not true feature unavailability as defined.
6. The notation system is overly complex and unclear. For example, while $P_i^{t}(x, y)$ is compared with $P_i^{ref}(x, y)$, the paper never defines how unavailable features are identified from actual data.
7. The rationale behind the selection of specific classes and clients for simulating “temporal feature unavailability” in the experiments is not provided.
8. The paper lacks ablation studies to verify the independent contribution of the proposed “2-step aggregation” compared with single-step aggregation.
9. The analysis of privacy and communication overhead is insufficient. Although the paper claims “minimal overhead,” it provides no quantitative comparison of communication volume, training time, or FLOPs.
10. While the authors discuss potential privacy-preserving mechanisms, no experiments are conducted to validate their effectiveness.

**Questions:**

1. How much of a decrease in Weight Update Variance triggers latent forgetting detection?
2. In the paper, $\cos(\theta) > 0$ is merely a sign judgment rather than a practical threshold. How is it selected during training? Moreover, how is this angle relationship measured or used to decide whether to continue using proxy samples?
3. Why are the latest FCL methods not included for comparison? MFCL is from 2023, yet the paper still labels it as “SOTA.”
4. In the sentence “the task is the same, the temporally or permanently unavailable features during training is usually unexpected and uncontrollable,” how should the “unavailable features” be specifically interpreted?

I look forward to the authors' rebuttal to address some of my remaining concerns.

---

> ### Author Response · Authors · 2025-11-24
> **Response to Official Review by Reviewer P3Tj (1/3)**
>
> We thank the reviewer P3Tj for recognizing the novelty of identifying the latent forgetting, our theoretical analysis, and providing constructive feedback that helps us improve our paper. We address each point in detail. Here is a brief summary:
>
> (1) We clarified the two types of latent forgetting.
>
> (2) We distinguished our principled, bias-correction Step-2 aggregation from generic fine-tuning.
>
> (3) We explained our choice of baseline and present new results with 2 more state-of-the-art methods.
>
> (4) We clarified the theoretical notation and meaning of unavailable feature groups and the rational for simulation of feature unavailability.
>
> (5) We justified that FedMemo preserves standard FL privacy and has comparable computational overhead with existing methods.
>
> (6) We demonstrated the reliability of using weight-update variance as a detection signal through optimizer variations and a stress test.
>
> (7) We explained the independent contribution of Step-2 aggregation.
>
> (8) We will improve the clarity of Figure 1.
>
> Below are detailed responses:
>
> **1.Clarification on types of latent forgetting:**
>
> Both types of latent forgetting come from the same cause: when some feature groups disappear, the aggregated gradient becomes biased toward the active ones. FedMemo handles both through WV-based detection and a gradient-correction step. We distinguish temporary and permanent cases because their behaviors differ—temporary unavailability causes brief collapses, while permanent loss creates a lasting bias—and baseline methods react differently in each scenario. Detecting short-term unavailability is especially important, since the correction is needed only during that period. We will add this clarification together with the corresponding learning curves in the revision.
>
> **2. 2-step aggregation vs fine tuning:**
>
> The reviewer is correct that our 2-step aggregation may resemble fine-tuning, but FedMemo is not an added fine-tuning phase. Fine-tuning optimizes the model on proxy data independently of aggregation and without considering how client gradients interact. In contrast, our Step-2 update is derived from the analysis in Section 3.3 (Propositions 3.2), which shows that latent forgetting comes from biased gradient weighting under temporal–spatial heterogeneity. The second step adds a targeted correction term -η(1-β)⟨Gref,Gp⟩ that reduces the reference loss when the synthetic proxy gradient aligns with the missing-feature direction and is applied only when the WV detector signals forgetting, to realign the global update with missing-feature directions. The mechanism is therefore a principled, detection-driven correction to the federated update rule, not generic proxy-based fine-tuning.
>
> **3. New baselines and why not more FCL baselines included:**
>
> We respectfully mention that our focus is on temporal-spatial heterogeneity in FL and though because it has been studied mostly in the FCL literature, we choose a strong baseline. Among existing methods, MFCL is the only recent work that directly models temporal–spatial heterogeneity and is the closest available baseline to our fully FL-based method.
>
> -We generally selected competitive baselines from three representative families of methods:
>
> (1) regularization-based approaches for spatial heterogeneity (**FedBR**),
>
> (2) GAN-based aggregation (**FedMGD**), and
>
> (3) temporal–spatial heterogeneity methods from the FCL literature (**MFCL**).
>
> FedBR [1] is a strong method to reduce the learning bias on local features and classifiers. It outperforms FedNDT, MOON, and FedMix, and FedMGD [2] is a GAN-based aggregation method outperforming both Gan-based and Distillation-based methods such as FedDF, FedGEN, and FedGAN.
> FedMemo constantly outperforms all baselines. New results are presented in the following tables.
>
> **Short-term latent forgetting:**
> | Method/ Dataset     | CIFAR10 (temporal) | CIFAR10 (final) | SVHN (temporal) | SVHN (final)
> |-|-|-|-|-|
> | FedBR    | 48.19    | 70.36      | 61.65  |  88.74 |
> | FedMGD    | 49.14   | 71.23      |  66.1  |  89.77 |
> |Client-side FedMemo|72.07 | 75.54| 90.96| 91.54|
> |Server-side FedMemo| 60.37| 78.5| 72.27 | 89.87|
>
> **Long-term forgetting:**
> | Method/ Dataset     | CIFAR10  | SVHN | CIFAR10 (CLP) | SVHN (CLP)
> |-|-|-|-|-|
> | FedBR    | 50.46    | 56.12      | 38.93  |  54.82 |
> | FedMGD    | 51.74   | 58.57      |  39.54  |  55.24 |
> |Client-side FedMemo|72.97| 91.02 | 26.7 | 52.19
> |Server-side FedMemo| 59.6 | 67.09 | 43.80 | 59.55
>
> **Gradual feature unavailability:**
> | Method/ Dataset     | CIFAR-10 (short-term) | CIFAR-10 (long-term) | SVHN (short-term) | SVHN (long-term)
> |-|-|-|-|-|
> | FedBR    | 70.02   | 68.73     | 85.23  |  83.20 |
> | FedMGD    | 70.94   | 69.24      |  86.42  |  84.11 |
> |Client-side FedMemo| 74.51| 74.07| 90.96 | 90.72|
> |Server-side FedMemo| 74.13| 72.2 | 89.42| 88.41

---

> > ### Author Response · Authors · 2025-11-24
> > **Response to Official Review by Reviewer P3Tj (2/3)**
> >
> > **4. Clarifying notation and the meaning of unavailable features, and rational for simulating feature unavailability (w6,w7,Q2)**
> >
> > In Section 3, the comparison between $p(x,y)$ and $p_i^{ref}(x,y)$ is used to formalize how temporal–spatial heterogeneity alters a client’s underlying joint distribution relative to a reference distribution. In this formulation, “unavailable features” correspond to coherent regions of the distribution that disappear over time—feature-correlated sample groups—rather than literal feature-vector dimensions. In high-dimensional vision data, such regions naturally align with class identities [3, 6], which is why class removal is a practical way to simulate the loss of specific feature groups. This modeling choice is also consistent with prior work (e.g., [3], [4],[6]), where the change in feature groups is simulated through labels.
> >
> > FedMemo does not attempt to identify unavailable features directly from raw data; instead, weight-update variance (WV) acts as a privacy-preserving signal indicating when these distributional regions have effectively vanished and induced gradient bias. The cosine-angle term in Proposition 3.2 serves only as a theoretical condition for when the correction step reduces the reference loss; it is not used as a threshold during training since the reference gradient is not accessible in FL. The WV detector is used in practice to trigger or stop the corrective step. We will revise the paper to clarify the relationship between the notation, the concept of unavailable features, and the practical detection mechanism.
> >
> > **5. On the privacy and computation costs:**
> >
> > Privacy:
> >
> > - FedMemo does not weaken FL privacy. No raw client data, labels, or features are ever shared.
> >
> > - Server-side FedMemo uses only server-generated synthetic proxy data.
> >
> > - Client-side FedMemo generates synthetic samples locally, and only standard model updates from real and synthetic training are sent.
> >
> > - WV detection is computed from aggregated updates already available in normal FL.
> >
> > Thus, FedMemo inherits the privacy guarantees of the underlying FL protocol (and can be combined with DP or secure aggregation without modification).
> >
> > Computation:
> >
> > Client-side FedMemo adds a small local cost from training on synthetic samples, while server-side FedMemo adds no client cost. Generator training happens once on the server(3.81 (s) on CIFAR10 and 3.72 (s) on SVHN), and synthetic-data generation on clients is negligible ((0.07 s for CIFAR10 and 0.06 seconds for SVHN).). The following table summarizes local training time compared to baselines (wall-clock seconds). Client-side FedMemo slightly increases the training time caused by training the synthetic data at the second step, but, as a trade-off, it can significantly improve performance by gaining 72.97% and 91.02 % accuracy on CIFAR10 and SVHN, while the next best performance is for MFCL which is 37.39% and 40.20% more expensive but falls short by 7.49% on 2.45% in accuracy, demonstrating our competitiveness.
> >
> > | Dataset/Method     | FedAVG | FedProx | SCAFFOLD | FedBR | FedMGD | MFCL | FedMemo
> > |-|-|-|-|-|-|-|-|
> > | CIFAR10   | 2.6   | 3.4     | 4.2  |  3.6 | 2.7 | 6.3 | 4.6 |
> > | SVHN   | 3.0   | 3.7     |  4.5  |  4.02 | 3.04 |  6.8 | 4.85 |
> >
> > Server-side FedMemo keeps client training cost identical to FedAvg, while client-side FedMemo adds moderate cost with significantly improved forgetting mitigation.
> >
> > **6. Reliability of WV as the Detection Signal for Latent Forgetting (added a stress test):**
> >
> > Our analysis and experiments show that weight-update variance (WV) separates latent forgetting from normal convergence dynamics. Under ordinary training, the magnitude and fluctuations are different than in latent forgetting with feature unavailability and the frequency of change is smooth. Latent forgetting produces a sustained, monotonic decline that is distinct from regular training. We further varied optimizers and learning-rate schedules and found WV dynamics different from latent forgetting. These results support WV as a robust and sufficient detection signal. In addition, we will add a **non-convergence stress test** (high learning rate): WV becomes clearly distinct from the decline observed during latent forgetting. ( We will also add WV curves for two other datasets **SVHN and GLUE** to the Appendix where the WV declines from approximately 20.62 to 7.10 on SVHN, and from 0.005 to 1.54e-5 on GLUE. )
> >
> > Detection does not rely on a fixed percentage drop. WV is treated as a time series, and the server applies a simple moving-average-with-threshold anomaly detector. We show that varying the threshold ratio (0.3–0.7 of the reference window) yields stable results with precision/recall/F1 ≈ 0.96 on CIFAR-10, and applying the anomaly detector of method in [5] achieves F1 = 98.2%. These results confirm that latent forgetting corresponds to a clear, statistically significant deviation in WV, not to ordinary convergence effects.

---

> > > ### Author Response · Authors · 2025-11-24
> > > **Response to Official Review by Reviewer P3Tj (3/3)**
> > >
> > > **7. Independent Contribution of Step-2 Aggregation**
> > >
> > > In FedMemo, Step-1 is the standard single-step FL update (same structure as standard FL) and by itself cannot correct the gradient bias that causes latent forgetting. Step-2 is injecting the missing-feature direction (derived from the proxy gradient) and actively mitigates forgetting.
> > >
> > > The comparison of step-2 already appears implicitly in our comparisons: all baselines use single-step aggregation only, especially FedMemo step-2 correction plus replay outperforms STOTA MFCL replay method. FedMemo’s consistent improvements over these single-step methods therefore highlights the independent contribution of Step-2.
> > >
> > > **8. Improving the Clarity of Figure 1:**
> > >
> > > We will improve Figure 1 illustration in the revision and make the difference between the two variants clearer.
> > >
> > >
> > > [1] Guo, Yongxin, Xiaoying Tang, and Tao Lin. "Fedbr: Improving federated learning on heterogeneous data via local learning bias reduction." International conference on machine learning. PMLR, 2023.
> > >
> > > [2] Sheng, Tao, et al. "Modeling global distribution for federated learning with label distribution skew." Pattern Recognition 143 (2023): 109724.
> > >
> > > [3]  Daiqing Qi, Handong Zhao, and Sheng Li. Better generative replay for continual federated learning. In Proceedings of the International Conference on Learning Representations (ICLR), 2023.
> > >
> > > [4] Anastasiia Usmanova, François Portet, Philippe Lalanda, and German Vega. A distillation-based ap proach integrating continual learning and federated learning for pervasive services. arXiv preprint arXiv:2109.04197, 2021.
> > >
> > > [5] Shreshth Tuli, Giuliano Casale, and Nicholas R Jennings. Tranad: Deep transformer networks for anomaly detection in multivariate time series data. arXiv preprint arXiv:2201.07284, 2022.
> > >
> > > [6] Babakniya, Sara, et al. "A data-free approach to mitigate catastrophic forgetting in federated class incremental learning for vision tasks." Advances in Neural Information Processing Systems 36 (2023): 66408-66425.

---

> > > > ### Comment · Reviewer_P3Tj · 2025-11-27
> > > > **State-of-the-art baselines**
> > > >
> > > > Thank you for the authors' response, which has addressed most of my concerns. However, I still remain skeptical about the authors' claim of using state-of-the-art baselines. To my knowledge, a number of recent studies have already investigated spatio-temporal heterogeneity, and it is unclear why these relevant methods were not included in the comparisons.

---

> ### Author Response · Authors · 2025-11-28
> **Addressing the concern about state-of-the-art baselines**
>
> We sincerely thank the reviewer P3Tj for highlighting the recent works. While they indeed address spatio-temporal heterogeneity in a quite different training procedures, we carefully excluded them because their fundamental problem settings and assumptions are incompatible with the specific task-free, privacy-preserving global modeling problem in Federated Learning (FL) that FedMemo solves. Below, we detail exactly why each method cannot serve as a valid baseline for our setting:
>
> 1. FedCroST [1]: fundamental difference in learning objective (personalized vs. global)
>
>         -  FedCroST is designed for personalized FL, where the goal is to fit distinct per-client trajectories rather than a single consistent global model. FedMemo, by contrast, aims to maintain a shared global model robust to missing features across all clients.
>
>         -  Privacy violation: FedCroST requires sharing local event representations and gradients, which violates the strict privacy constraints FedMemo adheres to (where no raw data or direct representations are shared).
>
> 2. STHFL[2] and STAMP [3]: incompatibility with task-Free / stage-Free settings of FL
>
>         -  STHFL relies on predefined “stages” of data arrival where samples accumulate. This is a stage-based formulation that cannot be relaxed into the standard stage-free FL setting FedMemo operates in. Additionally, STHFL assumes clients retain full feature coverage, whereas FedMemo is specifically designed to handle missing feature groups.
>
>         -  STAMP operates in a task-incremental setting with explicit task boundaries and requires access to past-task prototypes. This is incompatible with FedMemo’s or general FL’s task-free formulation, where data streams are continuous, task boundaries are undefined, and past data cannot be stored or revisited due to privacy and storage constraints.
>
> 3.  FedTA in [4]: inflexible assumptions on task progression
>
>         -   While addressing heterogeneity, FedTA’s global prototype fusion assumes stable task semantics where all clients follow the same task progression. It is not built to handle the dynamic, asynchronous shifts in data availability that FL has and targeted by FedMemo, nor does it support the relaxation of task boundaries required for FL.
>
> To summarize: we did not include these methods in experimental comparison because they solve different problems (personalization vs. global modeling) or rely on assumptions (stored prototypes, defined stages) that FedMemo explicitly aims to eliminate to suit FL requirements. Comparing against them would require altering their core mechanisms to fit our setting, rendering the comparison unfair or inconclusive. We will clarify these in the revised version of our manuscript.
>
> [1] Zhang, Yudong, et al. "Modeling spatio-temporal mobility across data silos via personalized federated learning." IEEE Transactions on Mobile Computing (2024).
>
> [2] Guo, Shunxin, et al. "STHFL: Spatio-Temporal Heterogeneous Federated Learning." arXiv preprint arXiv:2501.05775 (2025).
>
> [3] Nguyen, Minh-Duong, Le-Tuan Nguyen, and Quoc-Viet Pham. "Improving Generalization in Heterogeneous Federated Continual Learning via Spatio-Temporal Gradient Matching with Prototypical Coreset." arXiv preprint arXiv:2506.12031 (2025).
>
> [4] [3]Yu, Hao, et al. "Handling spatial-temporal data heterogeneity for federated continual learning via tail anchor." Proceedings of the Computer Vision and Pattern Recognition Conference. 2025.

---

### Official Review · Reviewer_1xHq · 2025-10-31

**Soundness:** 3
**Presentation:** 2
**Contribution:** 2
**Rating:** 4
**Confidence:** 4

**Summary:**

This paper introduces FedMemo, a novel framework to address the problem of "latent forgetting" in Federated Learning (FL). The authors identify and formalize a critical challenge in FL: temporal-spatial data heterogeneity, where features can become temporarily or permanently unavailable across clients and training rounds, leading to a feature-level "latent forgetting". FedMemo proposes a two-pronged approach: 1) a privacy-preserving detection mechanism for latent forgetting based on weight update variance, and 2) a two-step aggregation method that leverages synthetic proxy data (generated either on the server or clients by GAN) to mitigate the detected forgetting. The method is evaluated on both vision (CIFAR-10, SVHN) and language (GLUE) tasks, demonstrating significant performance improvements over strong baselines, including state-of-the-art methods adapted from Federated Class-Incremental Learning (FCL).

**Strengths:**

1. Clear motivation and presentation. The studied problem "temporal-spatial data heterogeneity" is challenging (but not novel) in the domain of federated continual learning.
2. Theoretical grounding: The paper provides a theoretical analysis (Propositions 3.1 and 3.2) that formally connects feature unavailability to performance degradation (latent forgetting) and explains how the proposed 2-step aggregation can mitigate it. This strengthens the methodological foundation.
3. Experimental results validate the better performance of the proposed approach in tackling the latent forgetting caused by temporal-spatial data heterogeneity, compared with several baselines and a state-of-the-art method.

**Weaknesses:**

1. Novelty is insignificant.
While the dynamic detection of latent forgetting in FL runtime is novel, the main idea for tackling the detected forgetting is replaying, which has been well-studied in the domain of (federated) continual learning.
2. The usage of the proposed FedMemo is not clear.
The paper proposes server-side and client-side FedMemo, but does not provide instructions on how to choose from these two variants. The necessity of client-side variant is not clear. This variant cannot ensure a superior performance in all FCL scenarios, compared with than the server-side variant. Additionally, it incurs extra computation burden on the clients.
3. Clarity of the 2-Step aggregation process needs further improvement.
While the high-level idea is clear, the exact sequence and timing of the two steps could be described more precisely in the main text. For instance, it is not clear how the server generates w_p^{t+1} in Equation (1) when the client-side mode is triggered. A more detailed algorithmic pseudo-code in the main paper would greatly improve clarity.
4. Experiment results are not illustrative.
(1) The effectiveness of client-side FedMemo is validated on only vision datasets. The authors do not provide any explanation on the missing results of client-side FedMemo on NLP tasks.
(2) The results supporting the effectiveness of weight update variance (WV) in detecting latent forgetting are reported only on one dataset (CIFAR-10). Therefore, it is hard to believe that the proposed WV is a general effective metric for detecting latent forgetting.
(3) The compared baselines are limited. Previous works have proposed a large number of FCL methods. Besides replaying-based methods, the authors should also test the other types of methods (e.g., regularization-based EWC, distillation-based CFeD, etc.) under the studied temporal-spatial data heterogeneity. See “Federated Continual Learning: Concepts, Challenges, and Solutions” for more details about the representative types of solutions to FCL and select more baselines from each type to compare.
(4) The experimental setups (e.g., the number and index of classes removed in different FL tasks) lack reasonability. Ablation studies with different experimental setups (e.g., setting the number of classes removed during learning to be 1,3,5,7, resepectively) are needed for validating the generality of the proposed FedMemo.
5. Extra computational overhead.
Although the paper argues that the client-side computation overhead is small in Appendix, a quantitative analysis (e.g., extra time or FLOPs compared to standard FedAvg) would make this claim more concrete, especially for resource-constrained edge devices.

Minor issues
- Typos: (1) “however,” in the line 115; (2) “MFCL Babakniya et al. (2023) and Yu et al. (2025) in FCL considers…, they assumes” in lines 257—258; (3) “and and” in the line 709; (4) “Client sideFedMemo” in Table 2; The format of “Client-side” and “Server-side” used in this paper is not unified. (5) “temporal-spatialheterogeneity,” in the line 408; (6) The sentence in the line 866 is not completed.

**Questions:**

1. Can you provide clear instructions on when to use the client-side FedMemo and when to use the server-side mode? Is there any cost-effective method to determine whether the clients’ generated synthetic data are more effective for tackling latent forgetting than the server’s generated data? Can the two modes be triggered simultaneously?
2. Once the 2-step aggregation is triggered, is it used until the end of learning, no matter whether the decreasing trend of WV is disappeared or not?
3. Why the results of "Client-side FedMemo" are not shown in Table 4? If client-side variant is not applicable to NLP tasks like GLUE, why do we need such a variant?
4. Can your proposed approach remain effective, if the experimental setup parameters (e.g., the number/indexes of removed classes) are changed?
5. Are the other types of FCL methods, like LWC, CFedD, applicable to your studied FCL scenarios (i.e., scenarios with temporal-spatial data heterogeneity)? If applicable, can you include them in the comparative experiments?

---

> ### Author Response · Authors · 2025-11-24
> **Response to Official Review by Reviewer 1xHq (1/3)**
>
> We thank the reviewer 1xHq for recognizing the challenges of the problem, our theoretical analysis, and our strong improvements over the state-of-the-art method. We appreciate the constructive feedback and address each point in detail. Here is a brief summary:
>
> (1) We explained the novelty of FedMemo by showing that it addresses temporal–spatial heterogeneity through a principled gradient-correction mechanism combined with a sophisticated data generation and reply method, rather than a simple replay method.
>
> (2) We provided clear guidelines on when to use server-side vs. client-side FedMemo and explain their complementary roles.
>
> (3) We clarified the 2-step aggregation rule and why client-side FedMemo is unnecessary for NLP tasks.
>
> (4) We explained client-side FedMemo for NLP tasks.
>
> (5) We added 2 more state-of-the-art methods in evaluation and demonstrated the clear performance advantage of our method.
>
> (6) We clarified the modeling of missing feature-groups, and added ablation study on different number of classes.
>
> (7)We explained why our weight-update variance detection approach is reliable and presented additional results for our weight-update variance detection approach, including a stress test.
>
> (8) Added computation overhead results.
>
> Below are detailed responses:
>
> **1. Clarifying the Novelty and Core Contributions of FedMemo:**
>
> While replay methods modify local training by mixing old and new data, our contribution lies in showing that temporal–spatial heterogeneity causes a biased global update and in designing a principled gradient-correction step, derived from this analysis, to counteract that bias during aggregation. FedMemo is different than existing replay methods, and outperforms existing STOTA replay methods in FCL, following is the list of our contributions:
>
> (1) Temporal–spatial heterogeneity in FL
>
> We study temporal–spatial heterogeneity in federated learning, where the feature groups in client’s data vary over time. Existing works generally assume fixed per-client distributions or predefined similar task boundaries for all clients in FCL and therefore do not analyze how such time-varying feature-group availability leads to latent forgetting during standard FL training.
>
> (2) A gradient-correction method based on theoretical analysis
>
> We show that temporal–spatial heterogeneity introduces a systematic bias in the gradients. From this analysis, we derive a two-step aggregation rule that adds a targeted correction to restore missing gradient directions. Unlike replay approaches—which modify client optimization by naively interleaving data—our method modifies the aggregation rule, correcting the update direction when some feature groups are unavailable.
>
> (3) An online detector for latent forgetting
>
> We provide a simple online detector based on weight-update variance that identifies when the aggregated gradient collapses (without using gradients) due to missing feature groups. This detector enables gradient correction to be applied only when needed. Existing methods do not include such runtime detection and typically rely on predefined task transitions.
>
> **2. When to Use Server-Side vs. Client-Side FedMemo (Response to W2 and Q1:)**
>
> Below, we provide clear guidance and a concise comparison.
>
> (a) Server-Side FedMemo:
>
> It is always safe to use, especially recommended during early training (critical learning periods (CLP) , and to detect CLP one can account on accuracy or use the metrics such as federated gradient norm (FGN) introduced in paper CriticalFL [1]). Server-side FedMemo is also the best choice when clients have limited computation resources.
>
> | Pros| Cons
> |-|-|
> | -Zero additional client cost|- Focuses on global correction rather than local corrections|
> |-Reliable even when the global model is unstable|  |
> |-Privacy-preserving|  |
> |-Consistently improves forgetting during all training phases|  |
> |- No dependency on client hardware|  |
>
> (b) Client-Side FedMemo:
>
> When clients have sufficient compute resources, and when the global model is stable, the generator is well suited to local distributions, client-side FedMemo enables improvements over server-side correction and provides additional gains.
>
> | Pros | Cons |
> |-|-|
> | - Adds client-specific correction | - Additional local computation (train on generated data)  |
> | - Often yields the best final accuracy in near-convergence stages  | - Not the best during CLP due to weaker generator quality (a limitation shared by all client-side generative methods, including MFCL) |
> | - Privacy-preserving: clients generate synthetic data locally from a shared generator trained on server |  |

---

> > ### Author Response · Authors · 2025-11-24
> > **Response to Official Review by Reviewer 1xHq (2/3)**
> >
> > Can Both Modes Be Used Together?
> >
> > Yes, in our experiments, we keep them separate for clarity, but the system can mix both if desired. One example of real world application is to use server-side and later switch to client-side if local resources are available. If don’t want to detect the CLP rounds or account on accuracy, or have limited local resources, use server-side FedMemo.
> >
> > **3. Clarity of 2-step aggregation :**
> >
> > FedMemo always checks WV at each round and continues applying 2-step aggregation **until the WV decreasing pattern continues**. This is important as the current methods, never discuss this problem and only make their settings predefined in CL and aware of time of data changings.
> >
> > Clarification on $w_p^{t+1}$ : In the client side FedMemo, each selected client performs a short synthetic train step to produce $w_{i,p}^{t+1}$, and the server aggregates them using the weighted average, producing a proxy update similar to Eq. (1). We will provide a clear step-by-step pseudocode in the revision for clarity.
> >
> > **4. Clarification of missing client-side FedMemo on NLP task:**
> > We don’t use client-side FedMemo for GLUE tasks. We use zero-shot generation in paper [2] to generate the synthetic data without access to any task-specific data. The zero-shot generator is not trained from the global FL model (does not depend on client updates). As a result, all synthetic text generation can be performed entirely on the server, and because the generator is not client-dependent, a client-side FedMemo variant offers no additional benefit for NLP. We will clarify this in the revision.
> >
> > **5. New baselines:**
> >
> > We select competitive baselines from three representative families of methods:
> >
> > (1) regularization-based approaches for spatial heterogeneity (FedBR),
> >
> > (2) GAN-based aggregation (FedMGD), and
> >
> > (3) temporal–spatial heterogeneity methods from the FCL literature (MFCL).
> >
> > **FedBR** [3]— a strong method to reduce the learning bias on local features and classifiers outperforms FedNDT, MOON, and FedMix, and **FedMGD** [4] — a GAN-based aggregation method outperforming both Gan-based and Distillation-based methods such as FedDF, FedGEN, and FedGAN.
> >
> > Because temporal–spatial heterogeneity has been studied mostly in the FCL literature, we include MFCL as the closest available baseline, even though our method is fully FL-based. MFCL requires task boundaries, which we relax when adapting it to FL for comparison.
> >
> > FedMemo constantly outperforms all baselines.
> >
> > **Short-term latent forgetting:**
> > | Method/ Dataset     | CIFAR10 (temporal) | CIFAR10 (final) | SVHN (temporal) | SVHN (final)
> > |-|-|-|-|-|
> > | FedBR    | 48.19    | 70.36      | 61.65  |  88.74 |
> > | FedMGD    | 49.14   | 71.23      |  66.1  |  89.77 |
> > |Client-side FedMemo|72.07 | 75.54| 90.96| 91.54|
> > |Server-side FedMemo| 60.37| 78.5| 72.27 | 89.87|
> >
> > **Long-term forgetting:**
> > | Method/ Dataset     | CIFAR10  | SVHN | CIFAR10 (CLP) | SVHN (CLP)
> > |-|-|-|-|-|
> > | FedBR    | 50.46    | 56.12      | 38.93  |  54.82 |
> > | FedMGD    | 51.74   | 58.57      |  39.54  |  55.24 |
> > |Client-side FedMemo|72.97| 91.02 | 26.7 | 52.19
> > |Server-side FedMemo| 59.6 | 67.09 | 43.80 | 59.55
> >
> > **Gradual feature unavailability:**
> > | Method/ Dataset     | CIFAR-10 (short-term) | CIFAR-10 (long-term) | SVHN (short-term) | SVHN (long-term)
> > |-|-|-|-|-|
> > | FedBR    | 70.02   | 68.73     | 85.23  |  83.20 |
> > | FedMGD    | 70.94   | 69.24      |  86.42  |  84.11 |
> > |Client-side FedMemo| 74.51| 74.07| 90.96 | 90.72|
> > |Server-side FedMemo| 74.13| 72.2 | 89.42| 88.41
> >
> > **6. Reasonability of experimental setup and ablation study on number/index of classes:**
> >
> > Removing classes is a standard and practical way to model missing feature-groups when explicit feature annotations are unavailable. This is consistent with FCL practice, where tasks are class-based but predefined and restricted. This modeling choice is also consistent with prior work (e.g., [5], [6], and MFCL), where the change in feature groups is simulated through labels, it is explicitly noted in [5]-- “ features are expected to be close to each other when their labels are the same”--, where feature groups are removed through predefined tasks boundaries.
> >
> > Following is the results for changing the number and index of removed classes for CIFAR10 dataset in short-term latent forgetting (the removed classes are selected randomly) where by removing more classes FedMemo remains effective.
> > | Method | 1 class | 3 classes | 5 classes | 7 classes|
> > |-|-|-|-|-|
> > | Client-side FedMemo | 74.61   | 74.43    | 74.04 |  73.96 |
> > | Server-side FedMemo    | 73.65   | 73.01      |  72.33  |  71.95 |

---

> > > ### Author Response · Authors · 2025-11-24
> > > **Response to Official Review by Reviewer 1xHq (3/3)**
> > >
> > > **7. Results for effectiveness and Robustness of WV :**
> > >
> > > We will add WV curves for two other datasets **SVHN and GLUE** to the Appendix where the WV declines from approximately 20.62 to 7.10 on SVHN, and from 0.005 to 1.54e-5 on GLUE.
> > >
> > > We have shown the robustness of WV when we varied optimizers and learning-rate schedules and found WV dynamics different than latent forgetting.  We also will add a **non-convergence stress test** (high learning rate) where WV becomes clearly distinct from the decline observed during latent forgetting and will add it to the revision.
> > >
> > > **8. Computational overhead:**
> > >
> > > (1) Client-side FedMemo computation time :
> > >
> > > In client-side FedMemo, the generator is trained on the server once the latent forgetting is detected, and is shared to clients. The one-time wallclock train time for the generator (server-side) is 3.81 (s) on CIFAR10 and 3.72 (s) on SVHN . The client-side wall-clock time for generating synthetic data is negligible (0.07 s for CIFAR10 and 0.06 seconds for SVHN).
> > > Local training results are detailed in the following table. Client-side FedMemo although slightly increases the training time caused by training the synthetic data at the second step it can significantly improve performance. It can significantly improve performance by gaining 72.97% and 91.02 % accuracy on CIFAR10 and SVHN, while the next best performance is for MFCL which is 37.39% and 40.20% more expensive but falls short by 7.49% on 2.45% in accuracy, demonstrating our competitiveness. . Following is the wall-clock (in seconds) local training time of FedMemo in comparison to baselines.
> > > | Dataset/Method     | FedAVG | FedProx | SCAFFOLD | FedBR | FedMGD | MFCL | FedMemo
> > > |-|-|-|-|-|-|-|-|
> > > | CIFAR10   | 2.6   | 3.4     | 4.2  |  3.6 | 2.7 | 6.3 | 4.6 |
> > > | SVHN   | 3.0   | 3.7     |  4.5  |  4.02 | 3.04 |  6.8 | 4.85 |
> > >
> > > -Server-side FedMemo computation time:
> > >
> > > In server-side FedMemo, the synthetic proxy data is trained on the server, so the local computations remain similar to FedAVG. The server trains the synthetic proxy data while the clients do the local training and performs the second step aggregation after receiving the updates and finishing training.
> > >
> > > [1]Yan, Gang, et al. "Criticalfl: A critical learning periods augmented client selection framework for efficient federated learning." Proceedings of the 29th ACM SIGKDD Conference on Knowledge Discovery and Data Mining. 2023.
> > >
> > > [2] Yu Meng, Jiaxin Huang, Yu Zhang, and Jiawei Han. Generating training data with language models: Towards zero-shot language understanding. Advances in Neural Information Processing Systems, 35:462–477, 2022.
> > >
> > > [3] Guo, Yongxin, Xiaoying Tang, and Tao Lin. "Fedbr: Improving federated learning on heterogeneous data via local learning bias reduction." International conference on machine learning. PMLR, 2023.
> > >
> > > [4] Sheng, Tao, et al. "Modeling global distribution for federated learning with label distribution skew." Pattern Recognition 143 (2023): 109724.
> > >
> > > [5]  Daiqing Qi, Handong Zhao, and Sheng Li. Better generative replay for continual federated learning. In Proceedings of the International Conference on Learning Representations (ICLR), 2023.
> > >
> > > [6] Anastasiia Usmanova, François Portet, Philippe Lalanda, and German Vega. A distillation-based ap proach integrating continual learning and federated learning for pervasive services. arXiv preprint arXiv:2109.04197, 2021.

---

> ### Comment · Reviewer_1xHq · 2025-11-24
>
> Thanks to the authors for the careful responses and additional experiment results. These additions have partially addressed my concerns. I would like to raise the score accordingly.

---

> > ### Author Response · Authors · 2025-11-28
> > **Official Response to reviewer 1xHq**
> >
> > We sincerely thank the reviewer 1xHq for careful reevaluation of our work and for raising the score. We are pleased to hear that our additional experiments and clarifications have addressed your concerns. If there are any remaining concerns or questions, please don’t hesitate to let us know and we are happy to answer.

---

### Official Review · Reviewer_VDS8 · 2025-11-03

**Soundness:** 2
**Presentation:** 2
**Contribution:** 2
**Rating:** 2
**Confidence:** 4

**Summary:**

The paper introduces the FedMemo framework, which detects forgetting using a weight-update variance signal computed from successive global updates, and mitigates it via a proxy-based two-step aggregation. The method is applicable on both server and client sides.

**Strengths:**

* articulation of temporal-spatial heterogeneity and why this yields latent forgetting.
* variance-based detection of forgetting.
* the pipeline is clearly described.

**Weaknesses:**

* several strong FL methods that mitigate forgetting and drift (e.g., Flashback, FeGAN, FL‑distillation approaches: FedNDT, FedDF) are mentioned in the related works but are not directly compared empirically.
* requires proxy data to train the server-side GAN (used 5000 samples). Client-side generator costs are also not measured.
* non-standard metrics for GLUE (accuracy for CoLA/MRPC/QQP), reduce comparability to prior work.
* the modeling of the temporal data heterogeneity excludes data points from a given client, which is not very realistic. Stronger work like Oort and REFL (which are not referenced) modeled client availability over time, which is a more realistic approach to vary the data distribution based on client availability.

**Questions:**

* does the proxy cover all classes, and with what class balance? What is the performance of a model trained only on the proxy (no FL)?
* in GLUE setup, clarify whether prefix parameters are applied only to the final layer or all layers, and why accuracies are low?

---

> ### Author Response · Authors · 2025-11-24
> **Response to Official Review by Reviewer VDS8 (1/2)**
>
> We thank the reviewer VDS8 for recognizing our findings on temporal–spatial heterogeneity and latent forgetting, and the effectiveness of our detection approach. We appreciate the constructive feedback and address each point in details below. Here is a brief summary:
>
> (1) We added 2 more state-of-the-art baselines (FedBR and FedMGD), and the new evaluation results showing our FedMemo consistently outperforms them.
>
> (2) We clarified the role and limitations of the small proxy dataset, show why it cannot replace the GAN, and report its standalone performance.
>
> (3) We provided detailed computation and generator-cost measurements for both server-side and client-side FedMemo.
>
> (4) We added more GLUE metrics and explained the prefix-tuning configuration.
>
> (5) We clarified why Oort and REFL are not appropriate baselines—these methods model client availability, not temporal–spatial feature-group unavailability within clients—which is the focus of our work.
>
> Below are detailed responses:
>
> **1. Adding new baselines:**
>
> **FedBR** [1]— a strong method to reduce the learning bias on local features and classifiers outperforming FedNDT, MOON and FedMix.
>
> **FedMGD** [2] — a GAN-based aggregation method outperforming both Gan-based and Distillation-based methods such as FedDF, FedGEN and FedGAN.
>
> Since the **temporal-spatial heterogeneity** is mostly studied in FCL literature, we originally choose **MFCL** as a strong baseline by adopting it to the standard FL.
>
> We didn’t choose Flashback as the baseline because it’s not a published paper and their implementation is not publicly available. We contacted the authors, but they were unable to provide their code.
>
> The following are our new results for new baselines across both CIFAR-10 and SVHN, FedMemo consistently outperforms both FedBR and FedMGD too.
> **Short-term latent forgetting:**
> | Method/ Dataset     | CIFAR10 (temporal) | CIFAR10 (final) | SVHN (temporal) | SVHN (final)
> |-|-|-|-|-|
> | FedBR    | 48.19    | 70.36      | 61.65  |  88.74 |
> | FedMGD    | 49.14   | 71.23      |  66.1  |  89.77 |
> |Client-side FedMemo|72.07 | 75.54| 90.96| 91.54|
> |Server-side FedMemo| 60.37| 78.5| 72.27 | 89.87|
>
> **Long-term forgetting:**
> | Method/ Dataset     | CIFAR10  | SVHN | CIFAR10 (CLP) | SVHN (CLP)
> |-|-|-|-|-|
> | FedBR    | 50.46    | 56.12      | 38.93  |  54.82 |
> | FedMGD    | 51.74   | 58.57      |  39.54  |  55.24 |
> |Client-side FedMemo|72.97| 91.02 | 26.7 | 52.19
> |Server-side FedMemo| 59.6 | 67.09 | 43.80 | 59.55
>
> **Gradual feature unavailability:**
> | Method/ Dataset     | CIFAR-10 (short-term) | CIFAR-10 (long-term) | SVHN (short-term) | SVHN (long-term)
> |-|-|-|-|-|
> | FedBR    | 70.02   | 68.73     | 85.23  |  83.20 |
> | FedMGD    | 70.94   | 69.24      |  86.42  |  84.11 |
> |Client-side FedMemo| 74.51| 74.07| 90.96 | 90.72|
> |Server-side FedMemo| 74.13| 72.2 | 89.42| 88.41
>
> **2. Regarding proxy data:**
>
> The small proxy dataset is not used for training, but only to initialize the GAN for generating synthetic data, using initial proxy is common in FL where limited public or proxy data are used for model tuning or hyperparameter selection. Thus, it does not compromise the privacy or realism of our setup. This proxy data has a balanced IID distribution but is too small to solve the main task. We trained the proxy separately on a central non-FL setting (using the same ResNe18 model as our experiments), and the accuracy of it on CIFAR10 and SVHN datasets are **34.26%** and **48.88%** which shows the imperfectness of the initial proxy data. We will add these training curves to the Appendix.
>
> **3. FedMemo computation time (including generator cost):**
>
> Client-side FedMemo generator and computation time: We respectfully clarify that the generator itself is trained entirely on the server only. Once the latent forgetting is detected, it is trained and shares its weights to clients. The one-time wallclock train time for the generator (on server) is **3.81 (s)** on CIFAR10 and **3.72 (s)** on SVHN. The client-side wallclock time for generating synthetic data is negligible (0.07 s for CIFAR10 and 0.06 seconds for SVHN). Local train time (in seconds) is detailed in the following table. Client-side FedMemo slightly increases the training time caused by training the synthetic data at the second step, but, as a trade-off, it can significantly improve performance by gaining 72.97% and 91.02 % accuracy on CIFAR10 and SVHN, while the next best performance is for MFCL which is 37.39% and 38.10% more expensive but falls short by 7.49% on 2.45% in accuracy, demonstrating our competitiveness.
>
> | Dataset/Method     | FedAVG | FedProx | SCAFFOLD | FedBR | FedMGD | MFCL | FedMemo
> |-|-|-|-|-|-|-|-|
> | CIFAR10   | 2.6   | 3.4     | 4.2  |  3.6 | 2.7 | 6.3 | 4.6 |
> | SVHN   | 3.0   | 3.7     |  4.5  |  4.02 | 3.04 |  6.8 | 4.85 |

---

> > ### Author Response · Authors · 2025-11-24
> > **Response to Official Review by Reviewer VDS8 (2/2)**
> >
> > Server-side FedMemo computation time: In server-side FedMemo, the synthetic proxy data is trained on the server, so the local computations remain similar to FedAVG. The server trains the synthetic proxy data while the clients do the local training, and performs the second step aggregation.
> >
> > **4. Standard metrics for GLUE datasets and clarification of the prefix tuning parameters:**
> >
> > We added more GLUE metrics: CoLA : Matthews correlation (Matt.), MRPC : F1-score,  QQP : F1-score. We still included average accuracy for completeness.
> >
> > For prefix tuning, we freeze all model parameters of the model and apply prefix tuning only to the final layer (mentioned in paper section 4), consistent with the design goal of keeping the parameter footprint minimal. Consequently, absolute accuracies are lower than those from full fine-tuning, which is expected because prefix parameters in a single layer provide limited adaptation capacity.  Nevertheless, FedMemo achieves the highest relative gains under the same setting (up to +25.69 % over FedProx), demonstrating its effectiveness in mitigating latent forgetting.
> >
> > | Method/ Dataset     | CoLA (Matt.) | MRPC (F1-score)|QQP(F1-score)
> > |---------------------------|-------------------------------|-------------------------------|--------------------------|
> > | Baseline (No feature unavailability)   | 32.23   | 90.26     | 86.07 |
> > | FedProx    | 3.21   | 0.0     |  0.0  |
> > |FedMemo| 35.12 | 89.16 | 85.74 |
> >
> > **Gradual feature unavailability:**
> > | Method/ Dataset     | CoLA (Matt.) | MRPC (F1-score)|QQP(F1-score)
> > |---------------------------|-------------------------------|-------------------------------|--------------------------|
> > | Baseline (No feature unavailability)   | 32.23   | 90.26     | 86.07 |
> > |FedProx| 28.15 | 42.23 | 78.01|
> > FedMemo | 33.22 | 90.37 | 85.84|
> >
> > **5. Modeling the temporal-spatial data heterogeneity, why not using OORT and REFL to vary the data distribution based on client availability?**
> >
> > Oort and REFL focus on modeling client availability, not changes in the data available within each client. Their goal is to select or schedule clients for resource efficiency, aiming to select clients to maximize training speed and efficiency while assuming that every client’s local distribution is fixed across time. In contrast, our work studies the missing of feature groups within a client and we don’t focus on resource heterogeneity. In FL deployments, these feature groups often vanish unpredictably: for example, a hospital may temporarily observe no samples of certain patient types. These situations require modeling changes inside each client’s distribution. This is aligned with FCL methods like [3],[4], and MFCL where feature groups are considered as labels as explicitly noted in [3]-- “ features are expected to be close to each other when their labels are the same”--, and feature groups are removed through predefined tasks boundaries.
> >
> > In image tasks, each class naturally represents a coherent feature-group distribution, so removing classes provides a practical and widely used way to simulate the disappearance of those feature groups over time capturing both spatial heterogeneity (different clients hold different groups) and temporal heterogeneity (a single client loses groups over time). In contrast, Oort and REFL keep each client’s data distribution fixed and vary which clients participate based on training speed or device capability, making them unsuitable baselines for studying latent forgetting caused by temporal–spatial feature-group unavailability. We will clarify this notion of feature-group unavailability in the revision.
> >
> > [1] Guo, Yongxin, Xiaoying Tang, and Tao Lin. "Fedbr: Improving federated learning on heterogeneous data via local learning bias reduction." International conference on machine learning. PMLR, 2023.
> >
> > [2] Sheng, Tao, et al. "Modeling global distribution for federated learning with label distribution skew." Pattern Recognition 143 (2023): 109724.
> >
> > [4]  Daiqing Qi, Handong Zhao, and Sheng Li. Better generative replay for continual federated learning. In Proceedings of the International Conference on Learning Representations (ICLR), 2023.
> >
> > [5] Anastasiia Usmanova, François Portet, Philippe Lalanda, and German Vega. A distillation-based ap proach integrating continual learning and federated learning for pervasive services. arXiv preprint arXiv:2109.04197, 2021.

---

> > > ### Author Response · Authors · 2025-11-28
> > > **Follow-up Regarding Reviewer VDS8's Comments**
> > >
> > > We sincerely appreciate your constructive feedback, which has helped us significantly in improving our work. We have carefully addressed all your comments in our response. Should you have any further questions or require additional clarification, please do not hesitate to let us know.

---

### Author Response · Authors · 2025-12-03
**Summary of Rebuttal**

Dear Area Chair,

Thank you in advance for your efficient handling of our submission under these exceptional circumstances. To assist in your assessment, we provide a concise summary of our rebuttal.

We have provided detailed responses to all reviewers, and please note that Reviewer 1xHq, zjdp, and P3Tj have already acknowledged that we have addressed most of their concerns and Reviewer VDS8 did not raise any new questions either. We highlight the key points addressed in the rebuttal below.

**- Added new baselines and new evaluation results requested by reviewers** (Reviewer VDS8 requested comparison with other FL baselines, Reviewer 1xHq requested methods beyond replay, and Reviewer P3Tj asked for inclusion of FCL baselines as comparison):

We incorporated FedBR and FedMGD and provided new results showing FedMemo consistently outperforms state-of-the-art regularization-based, GAN-based, and FCL-based temporal–spatial heterogeneity methods up to 7.49% on CIFAR10 and 6.46% on SVHN. On GLUE, FedMemo achieves up to 25.69% improvement over the best FL aggregation baseline, demonstrating substantial gains in NLP tasks too.

**- Clarified the novelty and mechanism** (Reviewer 1xHq asked for clearer differentiation from replay-style techniques, and Reviewer P3Tj questioned the difference of FedMemo from fine tuning):

FedMemo addresses latent forgetting arising from temporal–spatial heterogeneity via a principled gradient correction (Step-2) triggered by Weight-update Variance (WV) based detection. We clarified how FedMemo differs fundamentally from replay methods, which only mix local data without correcting biased gradients, and from fine-tuning, which optimizes on proxy data independently of aggregation and ignores how client gradients interact.

**- Demonstrated that our unique gradient correction mechanism allows us to bypass the need for high-fidelity proxies entirely** (validated by both theoretical analysis and empirical evaluation). (Reviewer zjdp requested clarification on the role of proxy and its quality impact in training the generator, and Reviewer VDS8 requested proxy performance):

We verified that FedMemo does not rely on high-quality proxy data using the following:

(i) Empirical evidence of poor proxy quality in our evaluation – achieves 34.26% on CIFAR-10 and 48.88% on SVHN when trained centrally on proxy data only.

(ii) Theoretical justification (coarse alignment). As detailed in our theoretical analysis (Section 3.3; Proposition 3.2), the Step-2 gradient correction only requires the proxy gradient to have a positive directional alignment (cosine similarity > 0) with the true gradient. It does not require the proxy samples to be high quality.

(iii) Robustness to GAN degradation. Sensitivity studies confirm robustness even when the GAN is deliberately degraded (50–70 rounds or 20% label noise).

**- Clarified server-side vs. client-side FedMemo, privacy and computation cost** (Reviewer 1xHq and zjdp asked for guidance on choosing between the two variants and privacy and together with Reviewer VDS8 requested the computational overhead):

We verified that FedMemo introduces no additional privacy risk beyond standard FL and clarified when to use server-side versus client-side FedMemo—including guidance on critical learning periods—and provided full computation and generator-cost measurements, showing minimal-to-moderate overhead while significantly outperforming all baselines.

**- Strengthened WV detection justification** (Reviewer P3Tj requested clarification on distinguishing forgetting from convergence in detection via Weight-update Variance (WV) and Reviewer 1xHq asked about generality across datasets):

We demonstrated WV’s reliability as a detection signal through training-dynamics variation experiments, additional high-LR stress test, and added SVHN/GLUE results confirming its generality across datasets.

**- Clarified modeling assumptions and baseline selection** (Reviewer VDS8 suggested OORT/REFL for modeling temporal-spatial heterogeneity, Reviewer 1xHq, P3Tj asked for rationale of modeling feature unavailability, and Reviewer P3Tj asked for inclusion of other temporal-spatial baselines):

We clarified why OORT/REFL are unsuitable (they model client availability, not feature-group unavailability), justified the rationale for modeling feature unavailability with ablations on the numbers of classes, and explained why recent temporal-spatial methods (FedCroST, STHFL, STAMP, FedTA) cannot serve as baselines for latent forgetting due to personalization, stage-based, or task-dependent assumptions and constraints.

We are confident that our response and revisions have thoroughly addressed the key concerns. Please feel free to contact us if any questions arise.

---

### Meta-Review · Area_Chair_yVk5 · 2026-01-06

**Summary:**

the authors propose FedMemo to address the challenge of latent forgetting in Federated Learning (FL). Unlike catastrophic forgetting in continual learning, latent forgetting occurs without explicit task boundaries. FedMemo introduces a privacy-preserving automated detection mechanism based to identify when forgetting occurs in real-time. Once detected, the framework triggers a proxy-based strategy using synthetic data to correct biased global gradients. The authors also perform various experiments to show the effectiveness of their method.
The reviewers found the problem of temporal-spatial heterogeneity in FL and providing a theoretically grounded solution interesting and liked the privacy-preserving nature of the approach. However, reviewers raised concerns about (1) lack of strong baselines, (2) high-quality data, and (3) unclear details, (4) lack of robustness of the metrics. While the authors response mitigated some of these concerns in my opinion the revision needed to the paper is still substantial and thus needs to go through another round of reviews.

**Reviewer Concerns:**

The authors partially addressed some concerns by adding new baselines and clarified aspect of proxy data. Still some recent baselines were excluded and the paper still relies even on low quality proxy data.

**Reviewer Scores:**

Reviewer 1xHq (Current: 4): Likely to increase to 5. The reviewer explicitly stated, "I would like to raise the score accordingly," after the authors provided the requested baseline comparisons. However initial objection by the reviewer was strong and many issues still remain.

Reviewer zjdp (Current: 2): Explicitly raised to 4. While the reviewer acknowledged improvements, they maintained that the fundamental assumption of having proxy data remains a limitation.

Reviewer VDS8 (Current: 2): Likely to stay the same. The authors addressed the reviewer's specific requests for GLUE metrics but it is unlikely IMO that this reviewer would increase their score given the initial review and response.

Reviewer P3Tj (Current: 4): Likely to remain 4. The reviewer acknowledged that most concerns were addressed but remained skeptical about the "state-of-the-art" claims.

---

### Decision · Program_Chairs · 2026-01-26

Reject